# Direct radiative forcing of light-absorbing carbonaceous aerosol and the influencing factors over China

Shuangqin Yang[1], Yusi Liu[2], Li Chen[1*], Nan Cao[1], Jing Wang[1], Shuang Gao[1]

[1]Faculty of Geography, Tianjin Normal University, Tianjin 300387, China

[2]State Key Laboratory of Severe Weather & Key Laboratory for Atmospheric Chemistry of China Meteorology Administration, Chinese Academy of Meteorological Sciences, Beijing 100081, China

*Correspondence to*: Li Chen (amychenli1981@126.com)

**Abstract.** Black carbon (BC) and brown carbon (BrC) are the dominant light-absorbing carbonaceous aerosols (LACs) that contribute significantly to climate change through absorbing and scattering radiation. This study used the GEOS-Chem

integrated with the Rapid Radiative Transfer Model for GCMs to estimate LACs properties and direct radiative forcings (DRFs) in China. Primary BrC (Pri-BrC) and secondary BrC (Sec-BrC) were separated from the organic carbon and modeled as independent tracers. The Chinese anthropogenic emissions of LACs emissions and the refractive indexes were updated. Additionally, we investigated the impacts of LACs properties and atmospheric variables on LACs DRFs based on principal component analysis. The results showed that the atmospheric annual mean clear-sky net DRFs of BC, Pri-BrC, and Sec-BrC

in China were $1.848 \pm 1.098$, $0.146 \pm 0.079$, and $0.022 \pm 0.008$ W m$^{-2}$, respectively. The atmospheric shortwave DRFs of BC and Pri-BrC were proportional to their corresponding concentrations, aerosol optical depth (AOD), and absorption aerosol optical depth (AAOD), and inversely proportional to single scattering albedo, surface albedo, and ozone concentration in most regions. The surface longwave DRFs for the LACs showed negative correlations with water vapor in most areas. The highest atmospheric warming effect of LACs was observed in Central China, followed by East China, owing to the high LACs

concentrations, AOD, and AAOD and low surface albedo and ozone concentration. Based on the net DRFs, we found that BC exerts a warming effect at the top of the atmosphere, while Pri-BrC and Sec-BrC induce a cooling effect. Within the atmosphere, they all can contribute to atmospheric heating, whereas at the surface, they collectively lead to surface cooling. This study enhances our understanding of the climatic impacts of LACs.

## 1 Introduction

The light-absorbing carbonaceous aerosols (LACs) generally contain black carbon (BC) and brown carbon (BrC) (Park et al., 2003; Jo et al., 2016 ). BC is the predominant absorber of LACs in visible wavelength with a weak spectrum dependence (Ramanathan and Carmichael, 2008; Hu et al., 2020) and can significantly impact atmospheric visibility (Bond et al., 2013; Cao et al., 2024). The global mean effective radiative forcing (RF) due to BC is 0.11 W m$^{-2}$ (IPCC, 2021), of which China accounts for 14% (Li et al., 2016; Zhou et al., 2024). BrC is a part of organic carbon (OC), which absorbs sunlight and has a

strong spectral dependence, especially at ultra-violet and short visible wavelengths (Kirchstetter et al., 2004; Laskin et al.,

2015; Saleh et al., 2015). BrC can influence ozone ($O_3$) photochemistry by strongly absorbing radiation, thereby reducing the photolysis rate of nitrogen dioxide ($NO_2$) (Jo et al., 2016; Wang et al., 2019). Furthermore, BC and BrC can influence climate change and atmospheric conditions by heating the atmosphere, dimming the ground surface, and melting the ice glaciers (Laskin et al., 2015).

BC is emitted into the atmosphere mainly by combustion processes from biomass burning, biofuel, and fossil fuel (Menon et al., 2002; Chung et al., 2012; Jo et al., 2016). BrC can be categorized into primary BrC (Pri-BrC) and secondary BrC (Sec-BrC). Pri-BrC can be emitted directly into the air through combustion along with BC, and its optical characteristics and chemical compositions vary with the fuel types and burning conditions (Pye et al., 2010; Laskin et al., 2015). Sec-BrC can be produced by the chemical conversion of gaseous precursors for instance volatile organic compounds (VOCs) emitted from

anthropogenic and biogenic sources (Pye et al., 2010). Previous studies based on models mainly focused on the direct radiative forcings (DRFs) of BC or Pri-BrC (Park et al., 2010; Feng et al., 2013; Saleh et al., 2015; Li et al., 2016; Mao et al., 2017; Wang et al., 2014, 2018; Carter et al., 2021; Zhang et al., 2021; Methymaki et al., 2023). Due to the complex source and light properties of Sec-BrC (Wang et al., 2018), few studies have considered the simultaneous absorption of BC, Pri-BrC, and Sec-BrC (Lin et al., 2014; Jo et al., 2016; Wang et al., 2014, 2018; Tuccella et al., 2020; A. Zhang et al., 2020). Although Xu et al.

(2024) estimated the DRF of Sec-BrC in China, their simulation was based on the global emission inventory with a resolution of $0.5° \times 0.5°$, and a unified BrC emission ratio was applied to all fuel types. The local emission inventory with higher resolution and more accurate model parameters should be improved. Furthermore, studies have shown that Sec-BrC DRF cannot be ignored as it is higher than Pri-BrC in the ultra-violet range in certain regions of China (Q. Zhang et al., 2020; Zhu et al., 2021). Therefore, the total BrC (Pri-BrC + Sec-BrC) DRF may be underestimated without the inclusion of Sec-BrC.

The estimation of LACs DRFs relies on the precise expression of LACs emission inventories and optical features (such as aerosol optical depth (AOD), absorbing aerosol optical depth (AAOD), single scattering albedo (SSA), and asymmetry factor) (Wang et al., 2014; Tuccella et al., 2020) as well as atmospheric variables (such as water vapor, $O_3$ concentration, and surface albedo) (Lu et al., 2020). The mass concentrations of LACs are usually calculated based on emission inventories and their optical properties can characterize the ability of LACs to absorb and scatter radiation. As atmospheric variables affect the

DRFs and heating rates caused by LACs, it is necessary to explore the relationships between the LACs DRFs and their influencing factors under different surface conditions.

Therefore, this study attempts to estimate the mass concentrations and optical properties of LACs over China using the GEOS-Chem chemical transport model; assess the DRFs of BC, Pri-BrC, and Sec-BrC based on local emission inventory; and determine the impacts of LACs properties and atmospheric variables on their corresponding DRFs in seven geographical

regions across China. Our study promotes knowledge of the climatic implications of LACs in China.

## 2 Data and method

### 2.1 Data

#### 2.1.1 Monitoring data

The monitoring data of LACs light absorption coefficients (Abs) were observed at 36 sites in China (Fig. 1) using an aethalometer (model AE-31, Magee Scientific Company Berkeley, USA) and used to determine the BC mass concentration based on the wavelength-dependent mass attenuation cross-section. There were five types of monitoring sites: three background sites, three remote sites, four rural sites, seven suburban sites, and 19 urban sites. Data were collected at seven wavelengths (370, 470, 520, 590, 660, 880, and 950 nm) during winter (January-February), spring (March-May), and summer (June-August) in 2017.

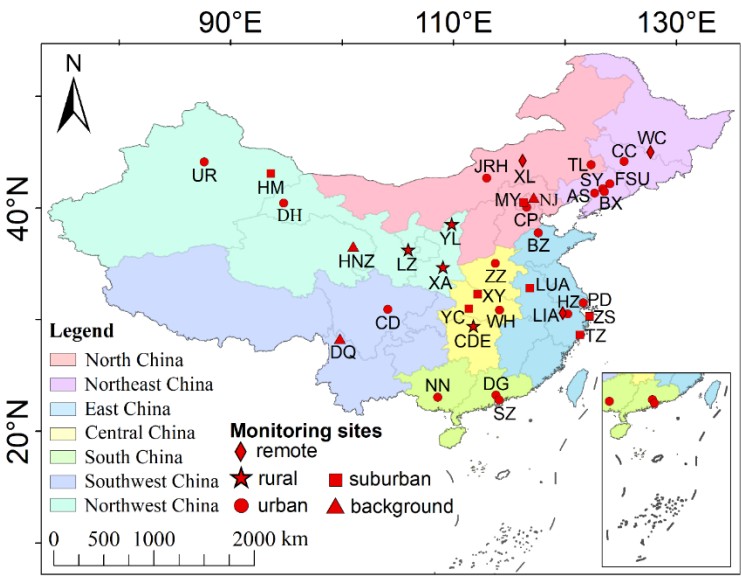

**Figure 1.** Location map of the 36 monitoring sites in China. Zhengzhou (ZZ), Zhoushan (ZS), Yulin (YL), Yichang (YC), Xiangyang (XY), Xilinhot (XL), Xi'an (XA), Wuhan (WH), Wuchang (WC), Urumqi (UR), Taizhou (TZ), Tongliao (TL), Shenzhen (SZ), Shenyang (SY), Pudong (PD), Nanning (NN), Nanjiao (NJ), Miyun (MY), Lanzhou (LZ), Lu'an (LUA), Lin'an (LIA), Jurh (JRH), Hangzhou (HZ), Hainan Tibetan (HNZ), Hami (HM), Fushun (FSU), Diqing (DQ), Dunhuang (DH), Dongguan (DG), Changping (CP), Changde (CDE), Chengdu (CD), Changchun (CC), Binzhou (BZ), Benxi (BX), and Anshan (AS).

The raw data from the aethalometer AE-31 model were corrected for further analysis in our study. The AE-31 instrument detects aerosol mass concentration and light intensity at seven channels with a time interval of 5 minutes and an airflow rate of 2 L min$^{-1}$. The light attenuation (ATN) through the quartz filter loaded with sampling aerosols can be calculated using Eq. (1) which was obtained from Weingartner et al. (2003).

$$ATN = -100 \times ln\left(\frac{I}{I_0}\right) \tag{1}$$

where I and $I_0$ denote the light intensity after passing the spot of the aerosol-laden filter and blank filter, respectively. Thus, the aerosol light absorption coefficients (Abs) can be calculated based on Eq. (2) (Weingartner et al., 2003).

$$Abs = \frac{\Delta ATN}{\Delta t} \times \frac{A}{Q} \times \frac{1}{C \times R(TAN)} \tag{2}$$

where $\Delta ATN$ is the variation of light attenuation within the time interval $\Delta t$; A is the area of the sampling filter spot (1.67 cm$^2$); and Q is the ambient air flow rate (2 L min$^{-1}$). C is an empirical correction factor (2.14), which is used to account for the quartz filter material and filter multiple scattering effects. R(ATN) is the empirical function to consider the filter shadowing effects, defined by Eq. (3) (Cao et al., 2024; Weingartner et al., 2003).

$$R(ATN) = \left(\frac{1}{f} - 1\right) \times \frac{ln(ATN) - ln(10)}{ln(50) - ln(10)} + 1 \tag{3}$$

where f represents the compensation parameter (Table S1), derived from (Rajesh and Ramachandran, 2018). The equivalent BC mass concentrations at a certain wavelength can be calculated based on the corresponding Abs and mass attenuation cross-section (MAC) using Eq. (4) (Weingartner et al., 2003).

$$[BC] = \frac{Abs}{MAC} \tag{4}$$

where MAC is an empirical calibration parameter to convert light absorption into aerosol concentration. The recommended MAC values for the AE-31 model are provided by the manufacturer and are listed in Table S1 (Rajesh and Ramachandran, 2018; Cao et al., 2024). A wavelength of 880 nm is regarded as the standard wavelength at which light absorption is attributed to BC (Weingartner et al., 2003; Zhou et al., 2024). Therefore, the equivalent BC mass concentration at 880 nm was adopted as the ground station concentration in our study.

By assuming the aerosol absorption is mainly contributed to BC and BrC, the separation of light absorption into BC and BrC follows the absorption Ångström exponent (AAE) segregation method, simply expressed by Eq. (5) and Eq. (6) as follows:

$$Abs_{BC}(\lambda) = Abs(880nm) \times \left(\frac{\lambda}{880}\right)^{-AAE_{BC}} \tag{5}$$

$$Abs_{BrC}(\lambda) = Abs(\lambda) - Abs_{BC}(\lambda) \tag{6}$$

where $Abs_{BC}(\lambda)$ and $Abs_{BrC}(\lambda)$ are the light absorption coefficients of BC and BrC at a given wavelength, respectively. AAE
indicates the wavelength dependence of aerosol and can be obtained by the fit of multi-wavelength absorption using the Power Law (Cao et al., 2024; Weingartner et al., 2003). The value of $AAE_{BC}$ is assumed to be 1.0, indicating "weak" spectral dependence of light absorption (Lack and Langridge, 2013; Cao et al., 2024). The light absorption coefficients for Pri-BrC and Sec-BrC are determined by the minimum R-squared approach, which is a BC-tracer method developed by Wang et al. (2019). More specific implementations of determinations for $Abs_{Pri-BrC}$ and $Abs_{Sec-BrC}$ can be found in Cao et al. (2024).

### 2.1.2 Aerosol Robotic Network (AERONET) data

The daily AOD and AAOD at 440 nm extracted from the Aerosol Robotic Network (AERONET) dataset used for model evaluations were taken from AERONET Version 3 cloud-screened Level 1.5 and quality-assured Level 2.0. The AERONET database contains long-term observations of ground-based global aerosol column AOD and AOD-dependent products, with AOD < 0.4 included in Level 1.5 but excluded in Level 2 (Chen et al., 2024).

### 2.1.3 Reanalysis data

The surface mass concentration, AOD, and AAOD at 550 nm for BC were obtained from the hourly M2T1NXAER dataset in the Modern-Era Retrospective Analysis for Research and Applications Version 2 (MERRA-2). MERRA-2 is the latest global reanalysis dataset produced by NASA's Global Modeling and Assimilation Office, with a horizontal resolution of $0.5° \times 0.625°$ (latitude $\times$ longitude) (Gelaro et al., 2017). The BC concentration (consisting of hydrophilic and hydrophobic BC) and BC AOD at 550 nm in the Copernicus Atmosphere Monitoring Service (CAMS) were also used in this study. CAMS is a global reanalysis dataset produced by the European Center for Medium-Range Weather Forecasts (ECMWF), with a spatial resolution of $0.75° \times 0.75°$ and temporal resolution of 3 h (Amarillo et al., 2024).

### 2.2 Atmospheric variables impacting LACs DRFs

Atmospheric variables, such as $O_3$ concentration in the troposphere, water vapor, and surface albedo, were included to explore their impacts on LACs DRFs. The monthly tropospheric column $O_3$ concentration with a spatial resolution of $1° \times 1.25°$ was obtained from the retrieval products of the Ozone Monitoring Instrument and Microwave Limb Sounder. It is determined via the tropospheric $O_3$ residual method, which subtracts the stratospheric column $O_3$ concentration observed by the Microwave Limb Sounder sensor from the total column $O_3$ concentration measured by the Ozone Monitoring Instrument sensor (Liang et al., 2024). The surface albedo with a spatial resolution of $0.1° \times 0.1°$ and monthly temporal resolution was derived from the fifth-generation road surface reanalysis dataset of ERA5-Land. This dataset can provide multiple variables for the water and energy cycles at the surface level and is distributed by the European Centre for Medium-Range Weather Forecasts (Muñoz-Sabater et al., 2021). The monthly water vapor with a horizontal resolution of $1° \times 1°$ was derived from the Moderate-resolution Imaging Spectroradiometer global-gridded Level 3 science product of MOD08_M3. The derivation of water vapor is based on an attenuation algorithm using observations from near-infrared channels (Gao and Kaufman, 2003; Chan et al., 2022).

### 2.3 Configuration of GEOS-Chem integrated with Rapid Radiative Transfer Model of General Circulation Model (RRTMG)

We employed the standard chemical transport model GEOS-Chem version 14.0.0, integrated with the Rapid Radiative Transfer Model for Global Circulation Model (RRTMG) in a configuration named GC-RT to simulate the LACs DRFs in China (Bey et al., 2001; Iacono et al., 2008; Heald et al., 2014). The GEOS-Chem model was driven by MERRA-2 assimilated meteorology data, created by NASA's Goddard Earth Observing System Version 5.12.4 (Gelaro et al., 2017). Our simulation was performed

at a 0.5° × 0.625° horizontal resolution in the nested domain of East Asia (60–150°E and 11°S–55°N), which covered the entirety of China; 47 hybrid sigma vertical layers stretching from the ground to the top of the modeled atmosphere (0.01 hPa, 80 km altitude), with the lowest level at about 60 m. The transport and chemistry time steps were set at five and 10 minutes, respectively (Carter et al., 2021). The required boundary conditions were derived from a 4° × 5° global simulation at time steps of 20 minutes for chemistry and 10 minutes for transport (Carter et al., 2021). The simulation was conducted from October 2016 to December 2017. The first three months were used for spinning to eliminate the impact of the starting conditions.

We conducted a standard tropospheric oxidant–aerosol simulation. Hydrophilic and hydrophobic carbonaceous aerosols were modeled separately, with hydrophobic BC and primary OC accounting for 80% and 50% of the initial emissions, respectively (Park et al., 2003). Hydrophobic aerosols then transformed into hydrophilic ones at an aging coefficient of nearly $10^{-5}$ s$^{-1}$ (Wang et al., 2018), indicating an e-folding time of 1.15 days (Cook et al., 1999). In terms of primary OC concentration, primary organic aerosol (OA) can be extrapolated by multiplying an OA/OC factor of 2.1, which stands for other non-carbon elements attached to OC (Aiken et al., 2008; Wang et al., 2018). Secondary OC, regarded as hydrophilic during our experiment, was generated by dynamic partitioning of semi-volatile products (Pye and Seinfeld, 2010), such as the oxidation of aromatic (Henze et al., 2008) and biogenic hydrocarbons (Pye et al., 2010). The wet removal scheme mainly followed the descriptions of Liu et al., (2001), including clearance by convective updrafts, rainout, and washout. The parameterization of dry deposition was based on a sequential resistance method introduced by Wesely (1989) and Zhang et al. (2001), which varies with local land-use types, meteorological situations, and particle size distribution. The global anthropogenic primary OC and BC emissions were derived from the Community Emission Data System version 2 (CEDS-2) inventory (Hoesly et al., 2018), with China's emissions were updated by the High-Resolution Integrated Emission Inventory of Air Pollutants for China (INTAC, 0.1° × 0.1°), which was developed by the team of Multi-Resolution Emission Inventory of China (MEIC) (Zheng et al., 2018; Wu et al., 2024)). Emissions from biomass burning sources were derived from the monthly Global Fire Emission Database Version 4 (GFED-4, 0.25° × 0.25°) inventory (van der Werf et al., 2017). The biogenic VOCs emissions were simulated online using the Model of Emissions of Gases and Aerosols from Nature Version 2.1 (MEGAN-2.1) scheme (Guenther et al., 2012). Emissions of mineral dust and aircraft were given by the Dust Entrainment and Deposition (DEAD) program (Zender et al., 2003) and the Aviation Emissions Inventory Code (AEIC) inventory (Simone et al., 2013; Stettler et al., 2011), respectively. Sea salt followed the accumulation and coarse scheme as described in Alexander et al. (2005) and Jaegle et al. (2011).

The LACs optical properties parameters were calculated using the standard Mie code by assuming aerosols were externally mixed with unimodal log-normal size distribution as presented in the Global Aerosol Data Set (Hess et al., 1998) and Drury et al., (2010). The enhancement of BC absorption by the coating was assumed to be 1.5 and the refractive index at 550 nm was adopted as 1.95–0.79i following the recommendation by Bond and Bergstrom (2006). The GEOS-Chem model was modified to calculate the AAOD and Abs for the entire column and the individual vertical layers. The output wavelengths used for computing the optical properties of the LACs were 370, 440, and 550 nm.

RRTMG calculates atmospheric radiation in the wavelengths between 0.23 and 56 μm, which can be further divided into 16 longwave (LW) bands and 14 shortwave (SW) bands (Iacono et al., 2008; Heald et al., 2014; Methymaki et al., 2023). It was

175 called every 3 h to compute the LACs DRFs at the top of the atmosphere (TOA), in the atmosphere (ATM), and at the surface (SUR) for LW, SW, and LW + SW (NET). The DRFs we output were at a daily time step, which were further averaged to obtain the annual mean values. The total flux and flux without a particular species can be calculated separately and the difference between the two fluxes is considered the DRF of interested species (Heald et al., 2014; Wang et al., 2014). Hence, the definition of aerosol DRF is as follows:

$$DRF_{NET} = DRF_{LW} + DRF_{SW} \tag{7}$$

$$DRF_{TOA}(LACs) = DRF_{TOA}(aerosol) - DRF_{TOA}(no_{LACs}) \tag{8}$$

$$DRF_{SUR}(LACs) = DRF_{SUR}(aerosol) - DRF_{SUR}(no_{LACs}) \tag{9}$$

$$DRF_{ATM}(LACs) = DRF_{TOA}(LACs) - DRF_{SUR}(LACs) \tag{10}$$

where DRF$_{NET}$, DRF$_{LW}$, and DRF$_{SW}$ are the direct radiative forcing of net, longwave and shortwave, respectively; TOA, SUR,
and ATM represent the top, bottom (surface), and middle of the atmosphere, respectively; aerosol denotes the case that contains all major aerosols, and $no_{LACs}$ means the absence of specific LACs aerosols.

## 2.4 Treatment of primary and secondary BrC

OC is regarded as a light-scattering aerosol in the default GEOS-Chem model. In this study, Pri-BrC and Sec-BrC were separated from the OC and modeled as independent tracers. Considering that a fraction of primary OC is BrC, we derived the
190 Pri-BrC emissions following the methodology presented by Jo et al. (2016), which is summarized as follows:

$$R_{BrC,EF} = R_{BrC,mass} \times \frac{EF_{BC}}{EF_{OC}} \tag{11}$$

$$R_{BrC,mass} = R_{BrC,abs} \times \frac{MAE_{BrC}}{MAE_{BC}} \tag{12}$$

$$(1 + R_{BrC,abs})(\frac{\lambda}{\lambda_0})^{-AAE_{LACs}} = R_{BrC,abs} \times (\frac{\lambda}{\lambda_0})^{-AAE_{BrC}} + (\frac{\lambda}{\lambda_0})^{-AAE_{BC}} \tag{13}$$

where $R_{BrC,EF}$ is the fraction of BrC in initially emitted OC; $R_{BrC,mass}$ is the BrC/OC mass ratio; $R_{BrC,abs}$ is the BrC/OC
absorption ratio, determined by linear regression analysis; EF is the emission factor, available from inventories; MAE is the mass absorption efficiency at the wavelength of $\lambda_0$ (550 nm). For more information about $AAE_{LACs}$, please see Jo et al. (2016). We referred to Zhang et al. (2021) to determine the above parameters using MAE$_{BC}$,550 = 7.50 m$^2$ g$^{-1}$, MAE$_{BrC}$,550 = 0.886 m$^2$ g$^{-1}$, AAE$_{BrC}$ = 5.48, and AAE$_{BC}$ = 1.0. For biomass burning emissions, $R_{BrC,EF}$ are displayed in Table S2; for biofuel emissions, we only considered sectors of energy, industry, and RCO (residential, commercial, and other) as contributors to
BrC, with $R_{BrC,EF}$ values of 0.468, 0.333, and 0.519, respectively. As the majority of light-absorbing secondary OC is related to aromatic hydrocarbons (Desyaterik et al., 2013; A. Zhang et al., 2020), aged aromatic secondary OC was treated as Sec-BrC.

In our simulation, the chemical and physical processes of Pri-BrC and Sec-BrC were consistent with those of the non-absorbing OC, except for their optical properties. The imaginary (k) and real parts of the aerosol refractive index denote the ability to absorb and scatter radiation (Laskin et al., 2015). Therefore, given that BrC absorption below 600 nm could not be ignored (Kirchstetter et al., 2004), the $k_{BrC}$ in this band scope was reallocated and the real part in this range and the complex refractive indexes in other ranges were kept in line with the corresponding non-absorbing OC. We calculated the primary $k_{BrC,550}$ using Eq. (14) derived from Sun et al. (2007), based on $MAE_{BrC,550} = 0.886$ m$^2$ g$^{-1}$ used in BrC emission calculation with the density of $\rho = 1.3$ g cm$^{-3}$; other k values were derived from Eq. (15) obtained from Saleh et al. (2014). The values of secondary $k_{BrC}$ (Table 1) were acquired from Wang et al. (2014), which was calculated based on $MAE_{BrC,440} = 0.3$ m$^2$ g$^{-1}$. Additionally, other optical properties of BrC, such as SSA and extinction efficiency, were recalculated based on Mie theory (Martin et al., 2003; Zhang et al., 2021).

$$k_{\lambda_0} = \frac{\rho\lambda_0 \times MAE_{\lambda_0}}{4\pi} \tag{14}$$

$$k_\lambda = k_{\lambda_0} \times \left(\frac{\lambda_0}{\lambda}\right)^{AAE-1} \tag{15}$$

**Table 1.** The imaginary refractive index of BrC ($k_{BrC}$) used in this study.

| λ (nm) | Pri-BrC $k_{BrC}$ | Sec-BrC $k_{BrC}$ |
|---|---|---|
| 300 | 0.211 | 0.050 |
| 400 | 0.112 | 0.023 |
| 440 | 0.088 | 0.014 |
| 500 | 0.065 | 0.006 |
| 550 | 0.050 | 0.004 |

**2.5 Principal component analysis (PCA)**

Principal component analysis (PCA) was used to explore the impacts of LACs properties (concentration, AOD, AAOD, and SSA) and atmospheric variables (surface albedo, $O_3$ concentration, and water vapor) on the LACs DRFs. PCA is a linear analysis tool that reduces dimensionality by explaining the total variance of the original data using fewer new variables known as principal components (Liu et al., 2023; Salazar-Carballo et al., 2024). Based on the PCA biplot, the correlation coefficient between any two variables can be identified by the cosine of the angle between them, with an angle lower than 90° indicating a positive correlation and higher than 90° indicating a negative correlation (Rencher and Christensen, 2012; Jolliffe and Cadima, 2016).

## 3 Results and discussion

### 3.1 General description of LACs and their light absorptions

Figure 2 shows a comparison between the simulated daily mean BC concentrations and the observed values at each site. The mean values of the correlation coefficient (R), root mean square error (RMSE), and normalized mean bias (NMB) at the 36 monitoring sites were 0.55, 2.47 µg m$^{-3}$, and -14%, respectively. The model simulations of BC concentrations at most sites were generally underestimated compared with the observed values. Figure S1 shows the time-series plots of daily BC concentrations derived from simulations and observations at each monitoring site. At 20 of the 36 sites, the daily trends in BC concentrations were characterized with R greater than 0.55. Figure 3 shows the spatial distribution of seasonal BC concentrations obtained from the simulations and reanalysis data. The magnitude of the simulated BC concentration was lower than that of the reanalysis data; however, the GEOS-Chem model captured the spatial pattern of the BC concentration. The BC concentrations displayed significant regional differences, with high levels in the east and low levels in the west. The mean simulated BC concentrations are 0.46 ± 0.30 µg m$^{-3}$ in spring, 0.31 ± 0.30 µg m$^{-3}$ in summer, 0.51 ± 0.37 µg m$^{-3}$ in fall, and 0.87 ± 0.60 µg m$^{-3}$ in winter. The increased LACs emissions from burning coal and biofuel for heating, reduced sunlight, lower temperature, and limited planetary boundary layer height all facilitate the accumulation of LACs at the surface (Peng et al., 2019; Feng et al., 2021; Liu and Tang, 2024; Cao et al., 2024), thereby leading to a peak in BC concentrations in winter. Increased precipitation during the summer monsoon can significantly enhance the wet deposition of LACs, resulting in low BC concentrations in summer (Cao et al., 2024).

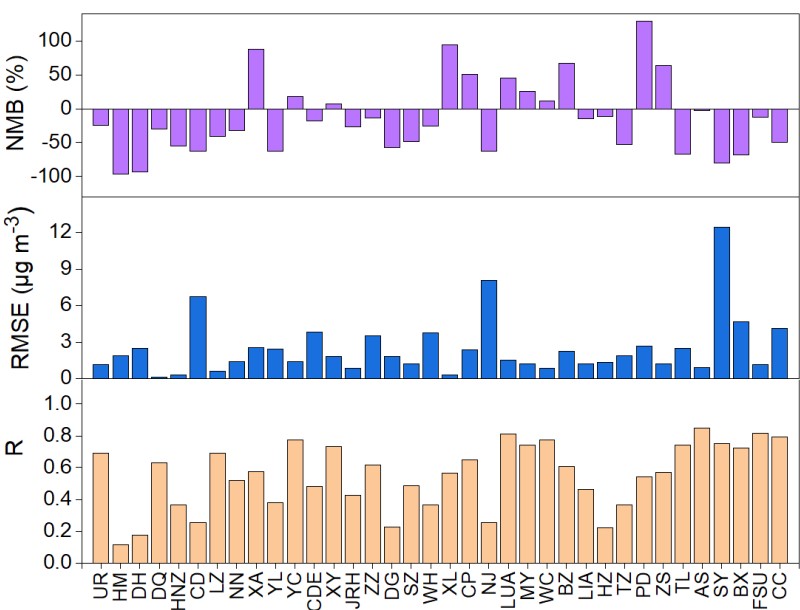

**Figure 2.** Comparisons of simulated versus observed daily mean BC concentrations at each site.

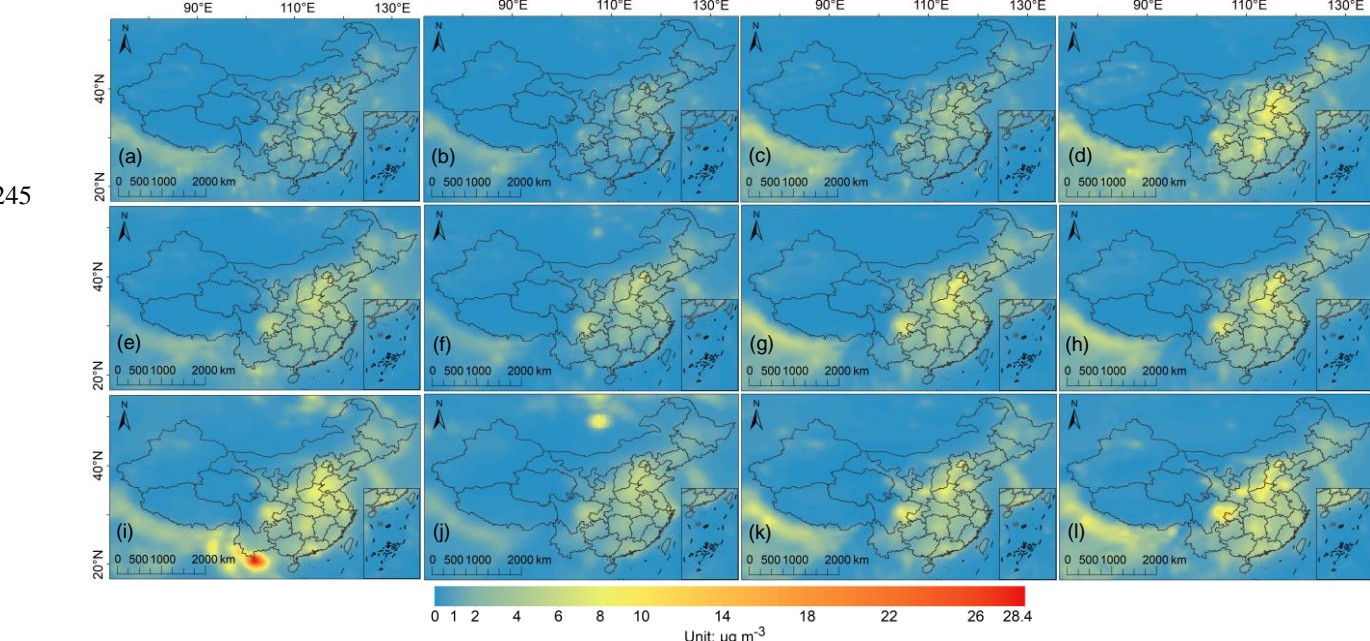

**Figure 3.** Spatial distribution of BC concentrations over China. Seasonal BC concentrations obtained from the GEOS-Chem simulation (a-d), MERRA-2 (e-h), and CAMS (i-l) in spring (a, e, and i), summer (b, f, and j), fall (c, g, k) and winter (d, h, l).

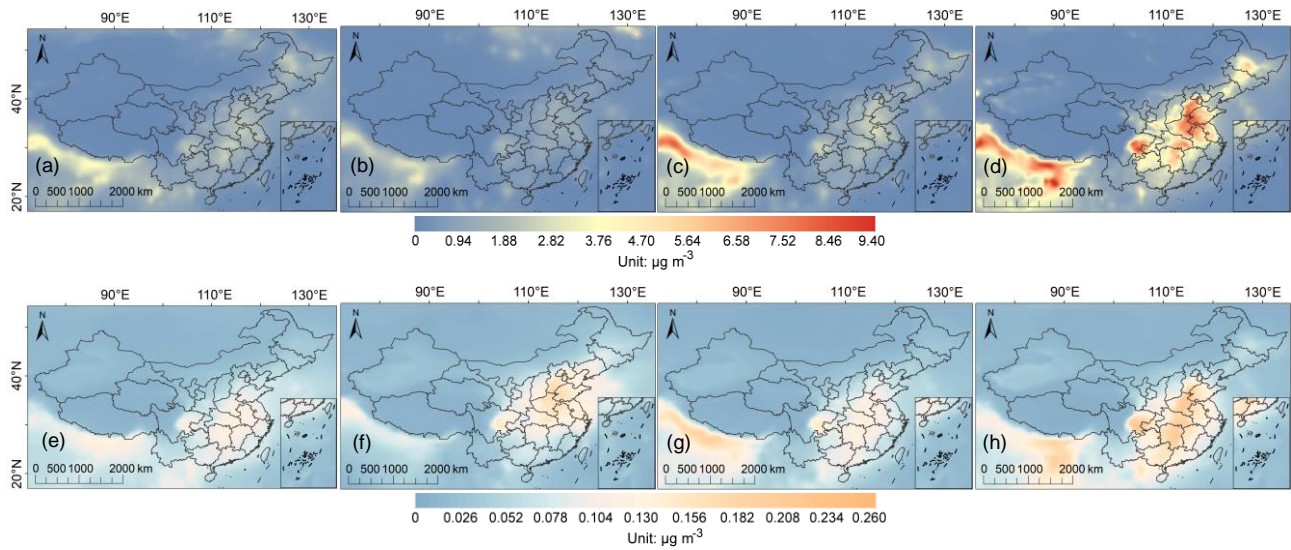

**Figure 4.** Spatial distribution of simulated concentrations for Pri-BrC and Sec-BrC over China. (a-d) seasonal Pri-BrC concentrations obtained from the GEOS-Chem simulation in spring (a), summer (b), fall (c) and winter (d). (e-h) same as (a-d) but for seasonal Sec-BrC concentrations.

Figure 4 shows the spatial distribution of simulated seasonal BrC concentrations over China for 2017. The high BrC concentration in the Sichuan Basin was mainly due to its unique topography, which leads to high relative humidity and low wind speed, thus beneficial to the accumulation of BrC (Peng et al., 2020; Cao et al., 2024). The GEOS-Chem model successfully captured the spatial variations of Pri-BrC and Sec-BrC, with both increasing from west to east, with similar results to those estimated by Jo et al. (2016). The seasonal mean Pri-BrC (Sec-BrC) concentrations were $0.51 \pm 0.30$ ($0.04 \pm 0.02$),

$0.35 \pm 0.35$ ($0.03 \pm 0.02$), $0.53 \pm 0.41$ ($0.03 \pm 0.02$), and $0.96 \pm 0.68$ ($0.05 \pm 0.02$) $\mu g\ m^{-3}$ for spring, summer, fall, and winter, respectively. The seasonal Sec-BrC concentrations were lower than those of Pri-BrC and the areas of high Sec-BrC concentrations were closer to southern China. The annual mean Sec-BrC concentration was $0.04 \pm 0.02\ \mu g\ m^{-3}$, which was probably at the low end of Sec-BrC concentration estimates, as the formation of Sec-BrC is only associated with aromatic secondary OC in our model.

Figure 5 shows the daily mean AOD and AAOD at 440 nm between the simulation and the AERONET data for the four sites in 2017. The R of AOD and AAOD were 0.76 and 0.70 for Level 1.5 and 0.81 and 0.64 for Level 2.0, respectively. This shows that our model captured both AOD and AAOD and reliably simulated aerosol optical properties. The NMB for AOD and AAOD at Level 1.5 was -33% and -28%, respectively, both slightly better than those of Level 2.0 products (-38% and -40%). The uncertainties associated with the assumptions about the physical properties of LACs within the GEOS-Chem model, such

as the exterior mixing state, log-normal distribution, and spherical morphology, can be propagated into the computations of the optical properties of LACs (Wang et al., 2018; Zhang et al., 2021), which may lead to discrepancies between the observed and simulated AOD and AAOD. Additionally, the AAOD product in the AERONET dataset may have uncertainties, especially in low AOD conditions (Dubovik et al., 2002). The accuracy level for SSA drops from 0.03 for AOD higher than 0.5 to 0.05–0.07 for AOD lower than 0.2 (Dubovik et al., 2002), which can introduce larger uncertainty into the final determination of

AAOD (AAOD = (1-SSA) × AOD). We further compared the simulated BC AOD (Fig. S2) and BC AAOD (Fig. S3) with the reanalysis data. The results indicated that the spatiotemporal patterns of AOD and AAOD for BC can be captured using GEOS-Chem.

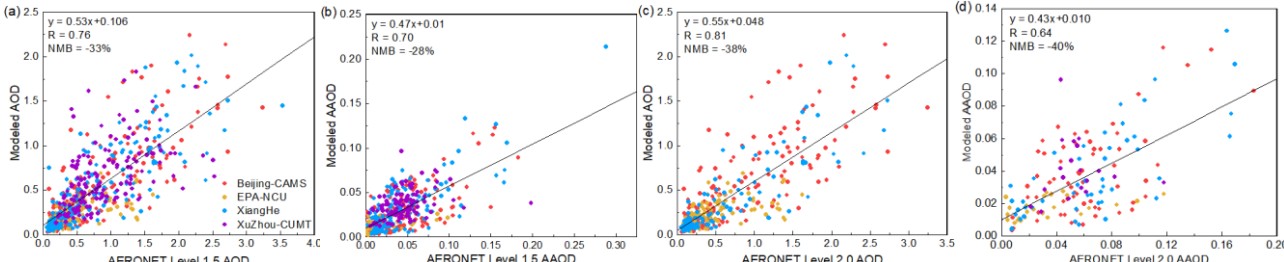

**Figure 5.** Comparison of simulated mean daily AOD (a, c) and AAOD (b, d) at 440nm with AERONET Level 1.5 (a, b) and 2.0 (c, d). Colored circles are the AERONET sites.

**3.2 Climate change contribution by BC and interactions with influencing factors**

Figure 6 shows the spatial distribution of the simulated annual mean clear-sky BC DRF in China. The values of the BC DRFs
for all-sky conditions (Fig. S4) were generally smaller than those for clear-sky conditions, which agrees with previous findings
(Feng et al., 2013; Heald et al., 2014; Lin et al., 2014). This can be primarily due to the scattering by clouds masking the
scattering by aerosols. The mean clear-sky BC DRFs across China for LW, SW, and NET at the TOA were $0.004 \pm 0.002$,
$0.381 \pm 0.238$, and $0.385 \pm 0.240$ W m$^{-2}$, respectively, whereas the surface mean values were $0.025 \pm 0.017$, $-1.487 \pm 0.883$,
and $-1.462 \pm 0.867$ W m$^{-2}$, respectively. A positive NET BC DRF at the TOA indicates a net heating effect on the top of the
atmosphere, whereas a negative value at the surface represents a net cooling effect on the ground. The positive NET BC DRF
within the atmosphere with a mean value of $1.848 \pm 1.098$ W m$^{-2}$ indicates that BC can strongly absorb solar radiation, leading
to atmospheric warming. BC warming effect is likely to be underestimated due to the ideal spherical shape of LACs
adopted in our model, which may lead to an overestimation of SSA by approximately 19-25% (Adachi et al., 2011;
China et al., 2015).

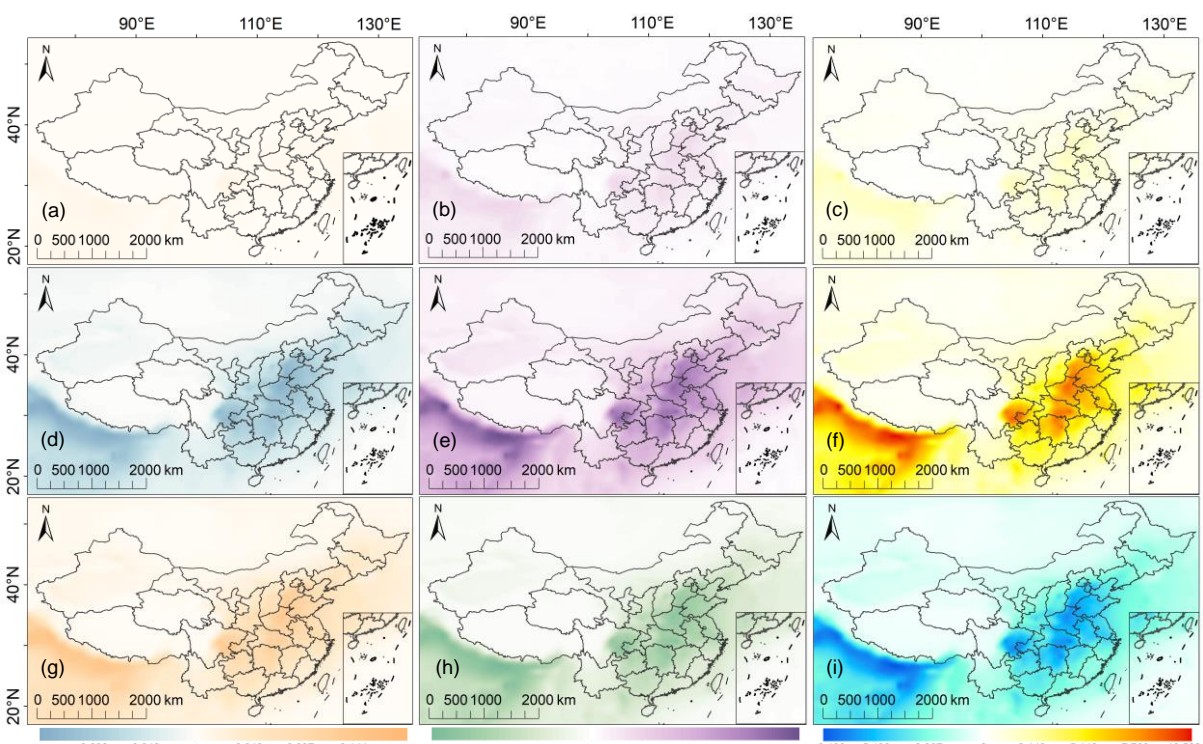

**Figure 6.** Annual mean clear-sky longwave (LW: a, d, g), shortwave (SW: b, e, h), and LW + SW (NET: c, f, i) DRF at the TOA (a, b, c),
in the ATM (d, e, f), and at the SUR (g, h, i) over China for BC.

Figure 7 shows the simulated monthly clear-sky BC DRFs for the seven regions in China. The highest LW BC DRF at the TOA was found in Central China, followed by East China (Fig. 7a), and the maximum values for these two regions occurred in March at 0.017 and 0.014 W m$^{-2}$, respectively. During winter and spring, the TOA SW BC forcing was higher in Central China than in other regions (Fig. 7b). The NET DRF of the BC was mainly determined by the SW DRF because the magnitude of the LW DRF was much smaller than that of the SW. Therefore, the trends of NET DRFs varied similarly to those of SW DRFs, with the strongest warming effect in Central China and the smallest in Northwest China. The surface LW and SW DRFs induced by BC (Fig. 7g and h) were significantly higher in Central, East, and South China than in the other four regions during winter, suggesting that the surface radiation in these three regions was more sensitive to light absorption from BC. Within the atmosphere, the SW BC DRFs in central-eastern and southern China generally showed a decreasing trend from January to July and an increasing trend until December, whereas the trends for the other four regions changed moderately (Fig. 7e).

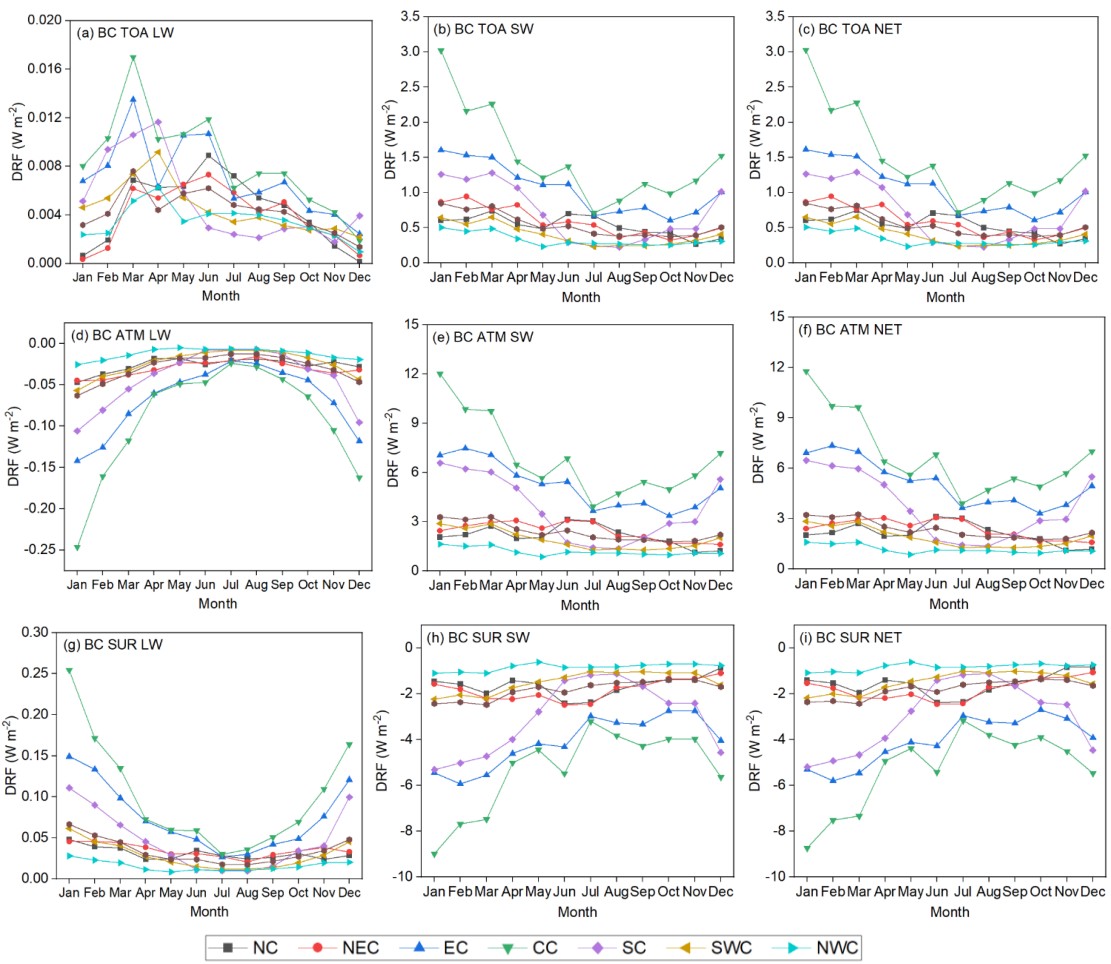

**Figure 7.** BC monthly mean longwave (LW: a, d, g), shortwave (SW: b, e, h), and LW + SW (NET: c, f, i) DRFs under clear-sky at the TOA (a, b, c), in the ATM (d, e, f), and at the SUR (g, h, i) over seven regions in China, including North China (NC), Northeast China (NEC), East China (EC), Central China (CC), South China (SC), Southwest China (SWC), and Northwest China (NWC).

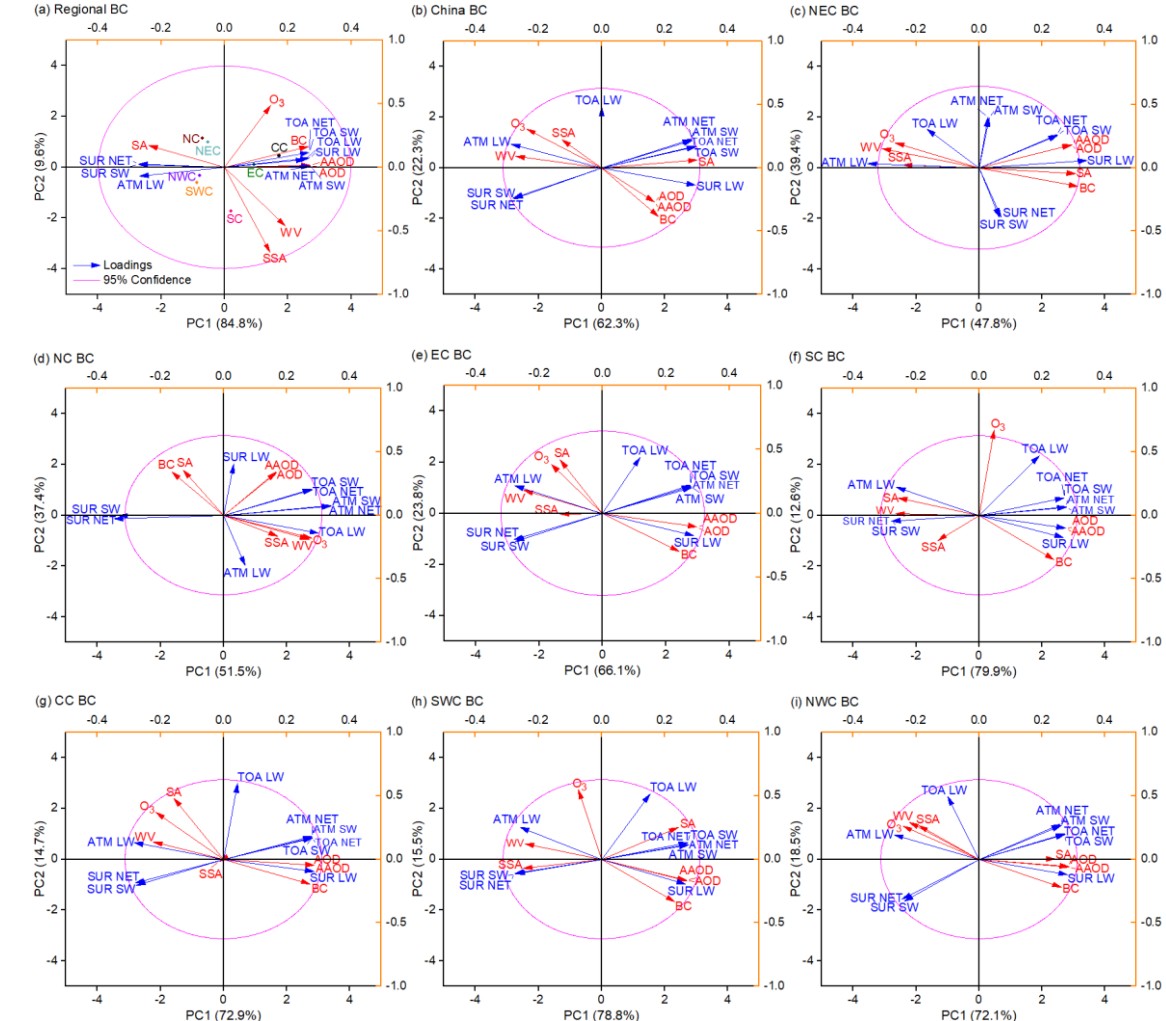

**Figure 8.** Principal component analysis biplots for BC DRFs and the influencing factors over seven regions in China (a), entire China (b), Northeast China (NEC, c), North China (NC, d), East China (EC, e), South China (SC, f), Central China (CC, g), Southwest China (SWC, h), and Northwest China (NWC, i). Blue and red arrows represent BC DRFs and their influencing factors, respectively. SA and WV are surface albedo and water vapor, respectively.

Figure 8 shows PCA biplots of the BC DRFs under clear-sky conditions versus the simulated BC properties and atmospheric variables. The principal components PC1 and PC2 for the BC PCA at the national scale (Fig. 8a) cumulatively explained more than 94% of the total variance, demonstrating that these indicators are strongly similar in East and Central China, whereas South China is different from the other six regions. Tables 2 and S5 present all the relationships between the BC DRFs and their influencing factors at the national and regional scales, respectively, with confidence levels within 95%. The angles between the atmospheric SW BC DRF and the BC concentration in most regions were acute, indicating a positive correlation. This is related to the absorption of SW radiation by the BC, which leads to opposite effects on the TOA and surface. There were two peaks in the ATM SW DRF in Central China, occurring in January and June. The peak in January was associated

with residential heating emissions and the peak in June was associated with an increase in BC concentrations (Table S4) from agricultural fires induced by the peak in agricultural activity for the harvest of winter wheat (seeded in the middle October and reaped in late May) (Huang et al., 2012; Ke et al., 2019). The differences between AAOD and AOD were so minor in the PCA biplots that they almost completely overlapped in all regions, suggesting that they exerted the same influence on the BC DRF.

The correlations between atmospheric SW DRF, AOD, and AAOD were positive for all regions. In contrast, BC SSA showed negative correlations with the corresponding atmospheric SW DRF in most regions. The aerosol SSA quantifies the fraction of scattering in the total extinction (scattering + absorption) (Bellouin et al., 2020). Thus, with increasing SSA, the absorption of solar radiation by BC decreases and scattering increases, which can lead to more solar energy being reflected into space and cooling the atmosphere. The acute angles between the atmospheric SW BC DRF and the surface albedo in South and Northwest China imply a positive relationship in these two regions. This is consistent with the theory that the higher the surface albedo, the higher the reflected solar radiation (Bibi et al., 2017; Lu et al., 2020; Chen et al., 2023), which creates more opportunities for BC to absorb SW radiation (Chen et al., 2024). The atmospheric SW forcing of BC was negatively correlated with $O_3$ concentration in all regions, except South, Northeast, and North China. The presence of $O_3$ can absorb SW sunlight (Xie et al., 2016), hence more SW radiation can be absorbed by increasing $O_3$ (Table S3). Consequently, the SW BC DRF in the atmosphere showed a decreasing trend during spring and summer (Fig. 7e). For all regions, the correlation between the LW BC DRF and water vapor was negative at the surface and positive within the atmosphere. These relationships are consistent with the principle that water vapor near the surface can strongly absorb LW radiation (Ramanathan and Coakley, 1978). Hence, the positive LW BC DRF at the surface (Fig. 7g) was at its lowest and the negative LW BC DRF in the atmosphere (Fig. 7d) was at its highest (less negative) during summer, when water vapor was at its maximum level (Table S3).

**Table 2.** Relationships between LACs DRFs and corresponding properties and atmospheric variables at a national scale. L, S, and N are LW, SW, and NET, respectively; Con, SA, and WV are concentration, surface albedo, and water vapor, respectively; "+" and "-" are positive and negative correlations respectively.

| LACs | BC | | | | | | | | | Pri-BrC | | | | | | | | | Sec-BrC | | | | | | | | |
|---|---|---|---|---|---|---|---|---|---|---|---|---|---|---|---|---|---|---|---|---|---|---|---|---|---|---|---|
| | TOA | | | ATM | | | SUR | | | TOA | | | ATM | | | SUR | | | TOA | | | ATM | | | SUR | | |
| DRFs | L | S | N | L | S | N | L | S | N | L | S | N | L | S | N | L | S | N | L | S | N | L | S | N | L | S | N |
| Con | - | + | + | - | + | + | + | - | - | - | - | - | - | + | + | + | - | - | - | + | + | - | - | - | - | + | + |
| AOD | - | + | + | - | + | + | + | - | - | - | - | - | - | + | + | + | - | - | - | + | + | - | - | - | + | + | + |
| AAOD | - | + | + | - | + | + | + | - | - | - | - | - | - | + | + | + | - | - | - | + | + | - | - | - | + | + | + |
| SSA | + | - | - | + | - | - | - | + | + | + | + | + | + | - | - | - | + | + | + | - | - | + | + | + | - | - | - |
| $O_3$ | + | - | - | + | - | - | - | + | + | + | + | + | + | - | - | - | + | + | + | - | - | + | + | + | - | - | - |
| SA | + | + | + | - | + | + | + | - | - | + | - | - | - | + | + | + | - | - | - | + | + | - | - | - | + | + | + |
| WV | + | - | - | + | - | - | - | + | + | - | + | + | + | - | - | - | + | + | + | - | - | + | + | + | - | - | - |

### 3.3 Climate change contribution by Pri-BrC and interactions with influencing factors

Figure 9 shows the spatial distribution of the simulated annual mean clear-sky Pri-BrC DRF in China. The spatial pattern of the Pri-BrC DRF showed high levels in central-eastern China and low levels in northwestern China. The mean LW Pri-BrC DRF at the TOA across China was $0.002 \pm 0.001$ W m$^{-2}$ with the maximum level in the Sichuan Basin. The slightly negative SW Pri-BrC DRF at the TOA ranged from -0.073 to -0.585 W m$^{-2}$, indicating a weak cooling effect of Pri-BrC at the TOA in our simulation. This may relate to the phenomenon that the cooling effects at longer wavelengths are greater than the warming

effects at shorter wavelengths within all SW bands (Arola et al., 2015). The mean TOA SW radiative forcing of Pri-BrC across China was $-0.251 \pm 0.098$ W m$^{-2}$, comparable to that of -0.220 W m$^{-2}$ obtained in Yao et al. (2017). The mean surface Pri-BrC DRFs in China at SW and LW are $-0.409 \pm 0.098$ and $0.014 \pm 0.098$ W m$^{-2}$ respectively. Thus, Pri-BrC produced a mean NET DRF of $-0.395 \pm 0.135$ W m$^{-2}$ at the surface, exerting a cooling effect on the ground. In addition, the cooling effect of Pri-BrC at the surface was almost twice that at the top of the atmosphere, largely owing to the absorption of SW sunlight by Pri-BrC

in the atmosphere. The atmospheric SW DRFs of Pri-BrC in China were positive, with a mean of $0.158 \pm 0.087$ W m$^{-2}$. The NET DRF for Pri-BrC within the atmosphere ranged from 0.031 to 0.350 W m$^{-2}$ and the mean value of $0.146 \pm 0.079$ W m$^{-2}$ was estimated to be 8% of that of BC, close to the 10% in Arola et al. (2015).

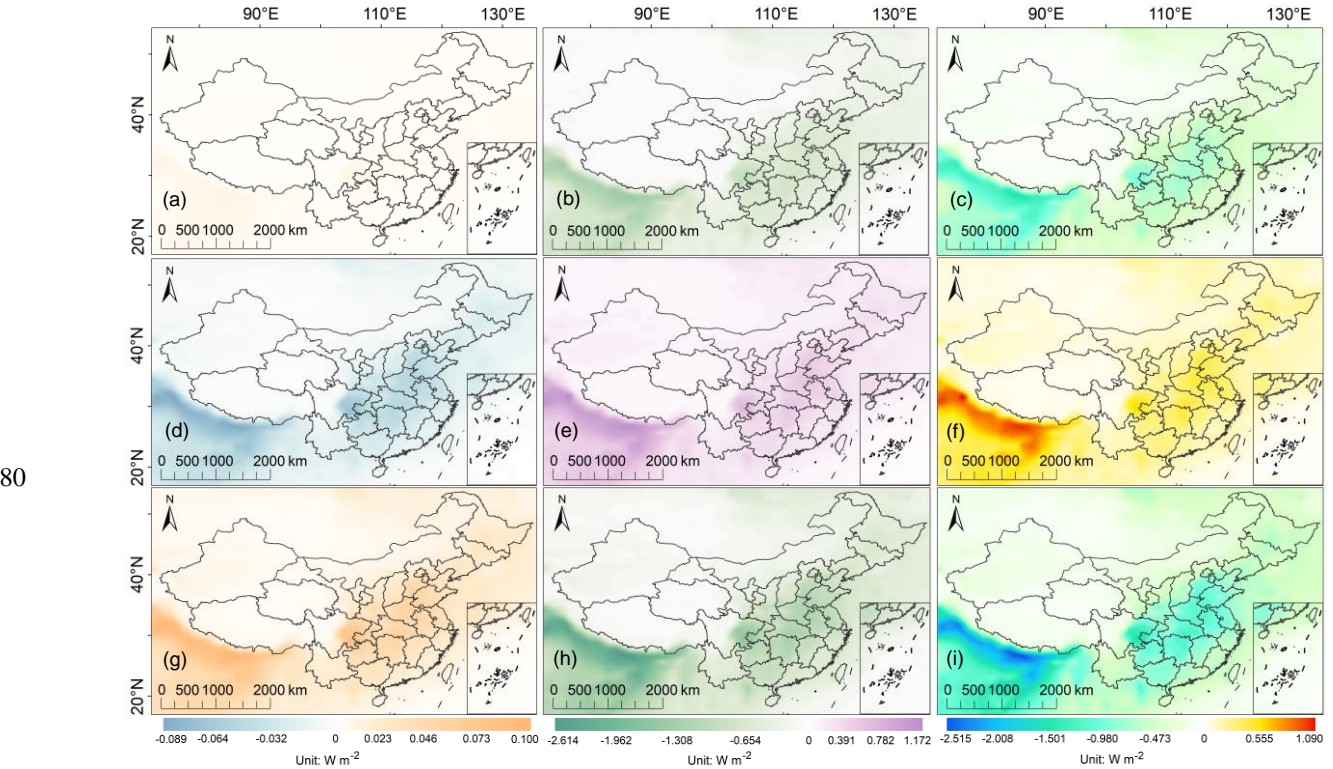

**Figure 9.** Annual mean clear-sky longwave (LW: a, d, g), shortwave (SW: b, e, h), and LW + SW (NET: c, f, i) DRF at the TOA (a, b, c), in the ATM (d, e, f), and at the SUR (g, h, i) over China for Pri-BrC.

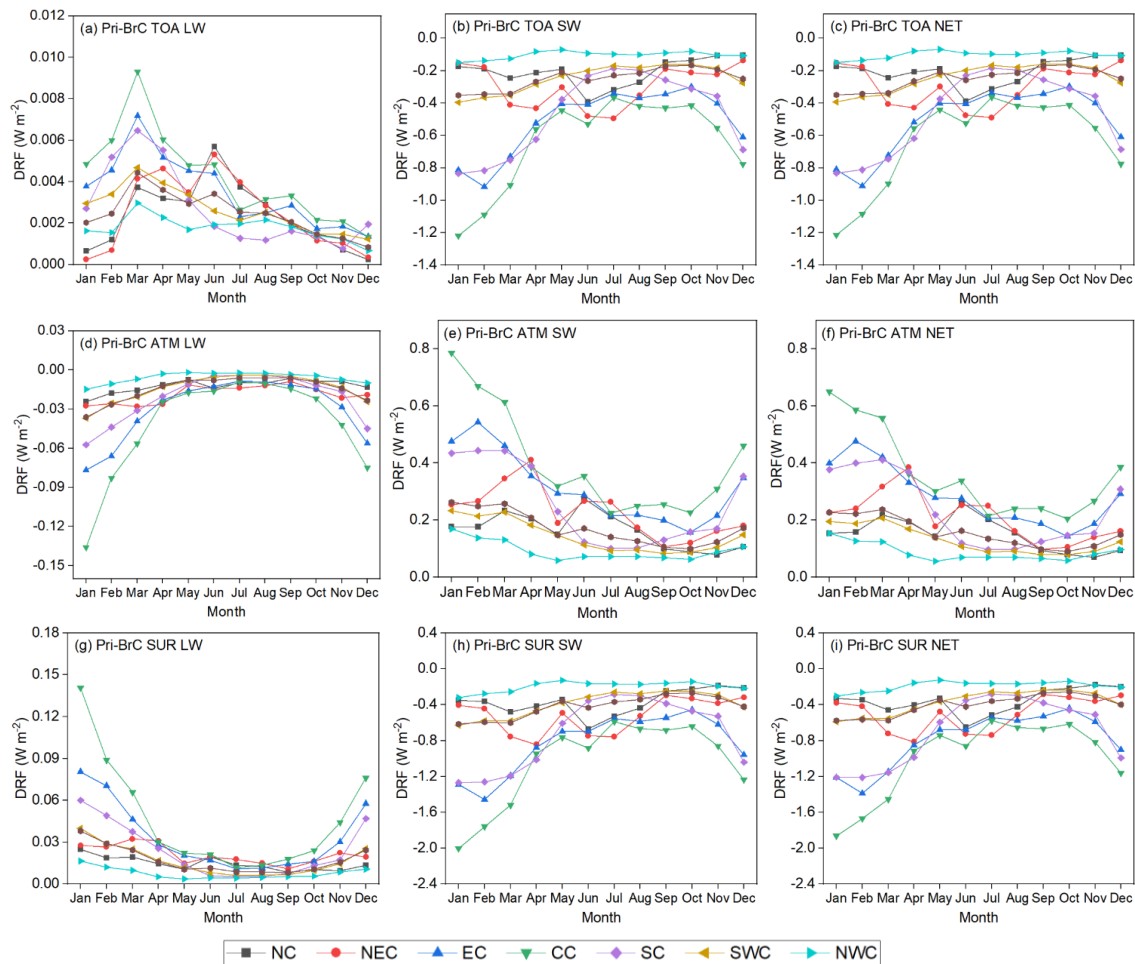

**Figure 10.** Pri-BrC monthly mean longwave (LW: a, d, g), shortwave (SW: b, e, h), and LW + SW (NET: c, f, i) DRFs under clear-sky at the TOA (a, b, c), in the ATM (d, e, f), and at the SUR (g, h, i) over seven regions in China, including North China (NC), Northeast China (NEC), East China (EC), Central China (CC), South China (SC), Southwest China (SWC), and Northwest China (NWC).

Figure 10 shows the simulated monthly clear-sky DRFs of Pri-BrC for seven regions across China. In contrast to other regions, the atmospheric SW DRF of Pri-BrC in Northeast China gradually increased from January to April and generally decreased until December, with a maximum value of 0.384 W m$^{-2}$ in April (Fig. 10e). The trends in the SW DRFs at the surface (Fig. 10h) were consistent with those at the TOA (Fig. 10b) for all regions but with greater values, suggesting that the absorption of SW by Pri-BrC exerted a relatively larger influence on the surface radiative balance in our simulation. The monthly trend of atmospheric NET forcing was most pronounced in Central China and lowest in Northwest China, with means of $0.362 \pm 0.154$ and $0.087 \pm 0.031$ W m$^{-2}$, respectively (Fig. 10f), and it was positive in all regions, suggesting that Pri-BrC is capable of warming the atmosphere.

Figure S5 shows PCA plots of the Pri-BrC DRF under clear-sky conditions versus the simulated Pri-BrC properties and atmospheric variables. PC1 and PC2 for Pri-BrC PCA at the national scale cumulatively explained more than 93% of the total variance and the similarities and differences in all variables between regions (Fig. S5a) were consistent with BC. The correlations between Pri-BrC DRFs and their influencing factors at national and regional scales are listed in Tables 2 and S7,

respectively. The SW DRF of Pri-BrC within the atmosphere in most regions was positively correlated with the corresponding concentration, AOD, and AAOD, and negatively correlated with the corresponding SSA. However, the minimum Pri-BrC SSA of 0.964 in all regions (Table S6) was significantly larger than the maximum BC SSA of 0.125 (Table S4), which may be another reason for the negative SW Pri-BrC DRF at the TOA. It may be related to the critical role of SSA in modifying the sign of aerosol DRF, with the critical SSA at 550 nm representing the shift of the DRF sign between positive and negative,

typically between 0.85 and 0.95 (Ramanathan et al., 2001; Li et al., 2022 ). The high SW DRF within the atmosphere during late winter and early spring in Northeast China (Fig. 10e) is attributed to the high Pri-BrC concentration and surface albedo (Tables S3 and S6). The high Pri-BrC concentration from January to March was caused by the combustion of coal for heating and the open biomass burning of agricultural residue to prepare for sowing spring wheat in the planting season (Ke et al., 2019; Wu et al., 2020). The mean surface albedo in Northeast China was $0.352 \pm 0.116$ during late winter and early spring and

remained around the minimum value of 0.155 from April to October before rising in November (Table S3). The $O_3$ concentration in most areas was negatively correlated with the atmospheric SW Pri-BrC DRF, similar to BC. The relationship between the LW Pri-BrC DRF and water vapor in all regions was similar to that of BC, indicating that water vapor exerted the same influence on the LW DRF for BC and Pri-BrC.

### 3.4 Climate change contribution by Sec-BrC and interactions with influencing factors

The spatial distribution of the simulated annual mean clear-sky Sec-BrC DRF in China is shown in Fig. 11. The spatial pattern of the Sec-BrC DRF resembled that of Pri-BrC, which showed high levels in eastern China and low levels in western China. The mean TOA LW DRF of Sec-BrC was $0.001 \pm 0.0003$ W m$^{-2}$ in China, with high levels in the Sichuan Basin and the North China Plain. The SW Sec-BrC DRF at the TOA ranged from -0.019 to -0.067 W m$^{-2}$ and the reason for the negative value was the same as Pri-BrC. The mean TOA NET Sec-BrC forcing in China was $-0.039 \pm 0.014$ W m$^{-2}$, which is of the same order of

magnitude as that simulated by Jo et al. (2016) for the global mean Sec-BrC DRF (-0.028 W m$^{-2}$). In addition, the mean SW DRF of total BrC at the TOA for China was $-0.290 \pm 0.079$ W m$^{-2}$, with a numerical sign identical to values from previous estimates (Shamjad et al., 2015, 2018; Zhang et al., 2021). The mean surface Sec-BrC DRFs at SW and LW in China were $-0.062 \pm 0.022$ and $0.002 \pm 0.001$ W m$^{-2}$ respectively. Thus, Sec-BrC exerted a cooling effect on the surface with a mean NET DRF of $-0.060 \pm 0.021$ W m$^{-2}$. Similar to Pri-BrC, the absolute value of the NET Sec-BrC DRF at the surface was almost twice

that at the top of the atmosphere, which was caused by Sec-BrC absorption of SW radiation. The SW DRF of Sec-BrC in the atmosphere was positive with a mean value of $0.023 \pm 0.008$ W m$^{-2}$. The atmospheric NET DRF for Sec-BrC ranged from 0.010 to 0.037 W m$^{-2}$. Compared to Pri-BrC, the areas of high DRFs for Sec-BrC were more extensive and closer to the southern regions of China, where the atmospheric NET DRFs were greater than 0.034 W m$^{-2}$. The mean NET Sec-BrC DRF

in the atmosphere for China was $0.022 \pm 0.008$ W m$^{-2}$, which indicates that Sec-BrC contributes to atmospheric warming. In addition, the DRFs of Sec-BrC were smaller than those of Pri-BrC, suggesting that the atmospheric absorption of solar radiation by Sec-BrC was weaker than that of Pri-BrC. The radiative forcing of Sec-BrC is likely to be underestimated. This may result from the utilization of idealized Sec-BrC physical parameters and the exclusive consideration of aromatic secondary OC as BrC.

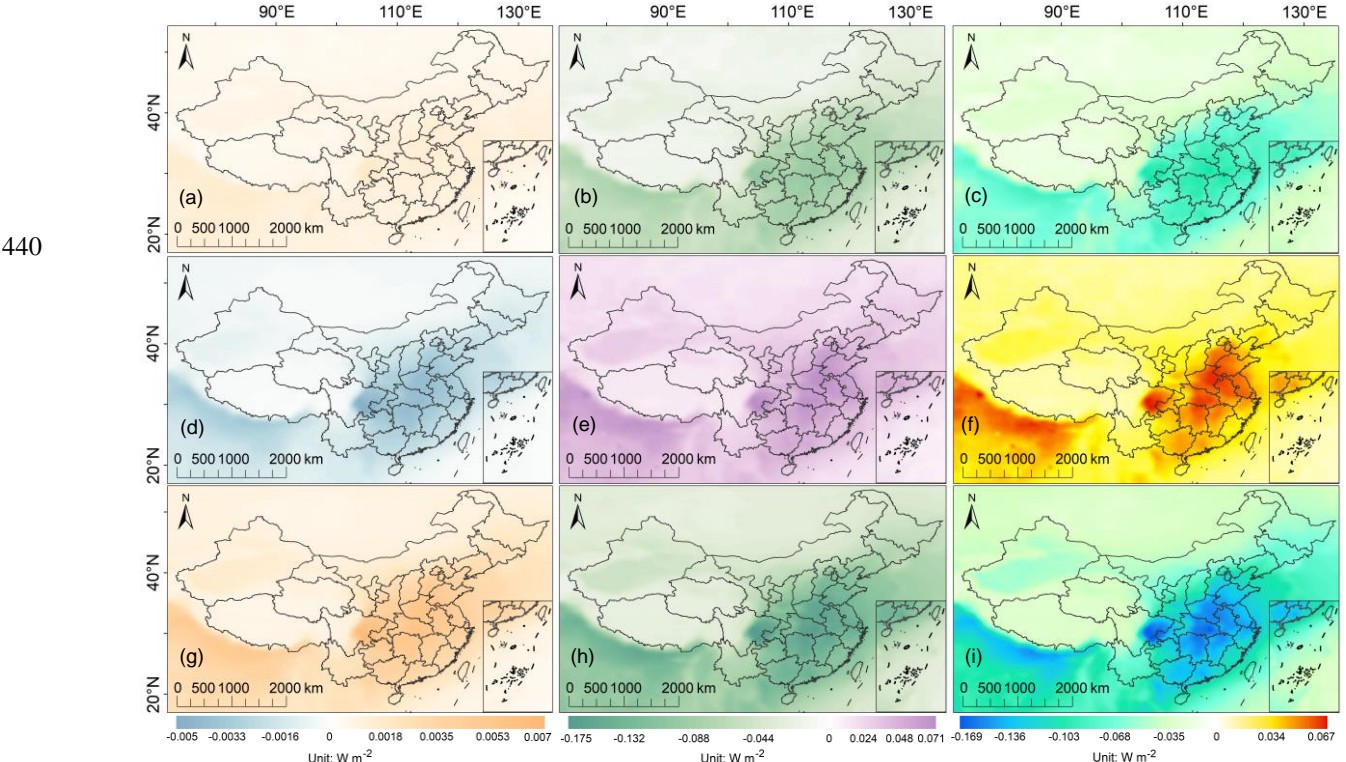

**Figure 11.** Annual mean clear-sky longwave (LW: a, d, g), shortwave (SW: b, e, h), and LW + SW (NET: c, f, i) DRF at the TOA (a, b, c), in the ATM (d, e, f), and at the SUR (g, h, i) over China for Sec-BrC.

Figure 12 shows the simulated monthly clear-sky DRFs of Sec-BrC for seven regions across China. The TOA LW forcings for Sec-BrC were slightly positive in all regions, increasing gradually from January through September and decreasing through December (Fig. 12a). As absolute values of the SW Sec-BrC DRFs at the TOA (Fig. 12b) were smaller than those at the surface (Fig. 12h), the atmospheric SW Sec-BrC DRFs were positive in all regions (Fig. 12e), showing an increasing trend from late winter to summer and subsequently decreasing until December except in South China.

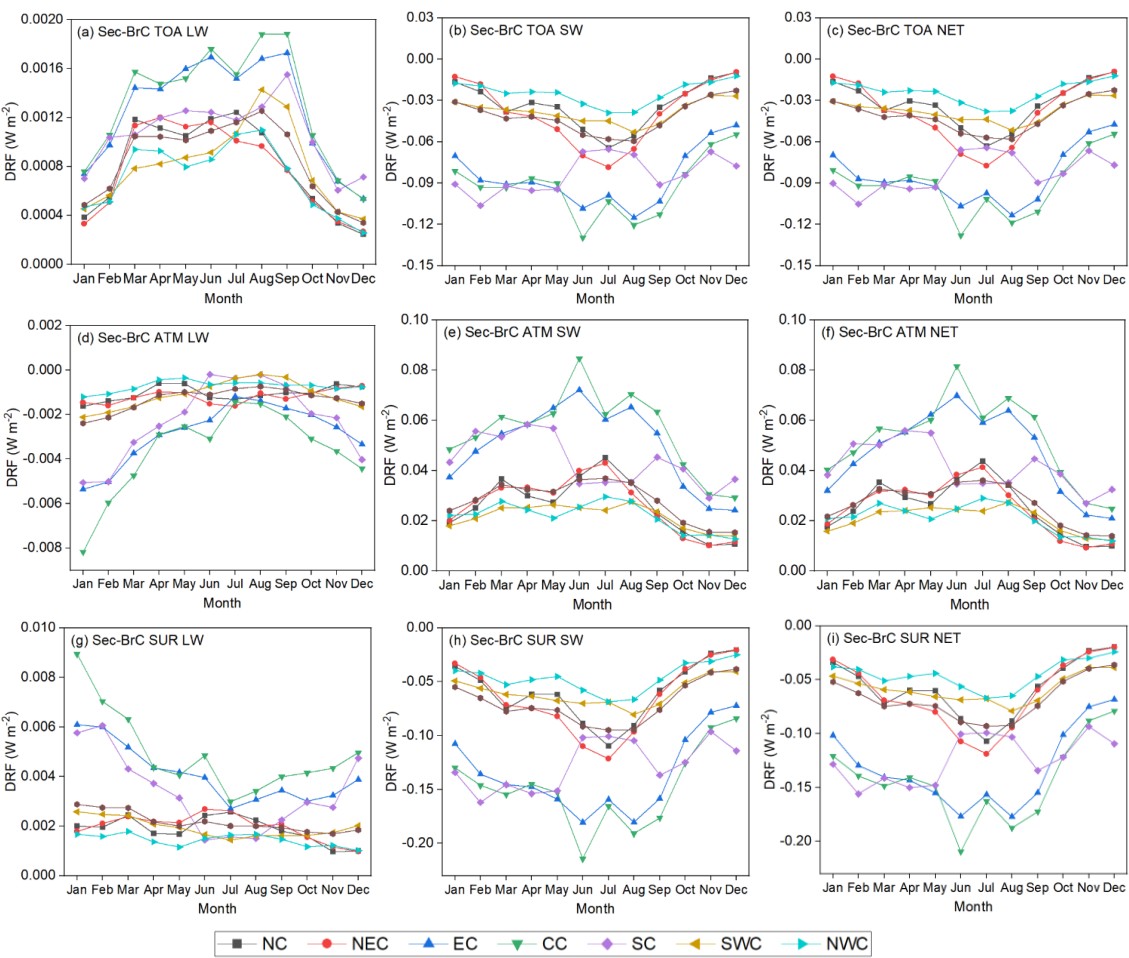


**Figure 12.** Sec-BrC monthly mean longwave (LW: a, d, g), shortwave (SW: b, e, h), and LW + SW (NET: c, f, i) DRFs under clear-sky at the TOA (a, b, c), in the ATM (d, e, f), and at the SUR (g, h, i) over seven regions in China, including North China (NC), Northeast China (NEC), East China (EC), Central China (CC), South China (SC), Southwest China (SWC), and Northwest China (NWC).


Figure S6 shows PCA plots of the Sec-BrC DRF under clear-sky conditions versus the simulated Sec-BrC properties and atmospheric variables. PC1 and PC2 for the Sec-BrC PCA at the national scale cumulatively explained more than 97% of the total variance and all indicators involved in the Sec-BrC PCA procedure showed a similar pattern of regional clustering as Pri-BrC (Fig. S6a). The correlations between Sec-BrC DRFs and their influencing factors at national and regional scales are listed

in Tables 2 and S9, respectively. The atmospheric SW DRF of Sec-BrC in most regions was positively correlated with the corresponding concentration, AOD, and AAOD. The influences of Sec-BrC concentration, AOD, and AAOD on Sec-BrC atmospheric SW DRF were positive, similar to that of BC and Pri-BrC. The relatively high concentration of Sec-BrC in summer (Table S8) may be related to the phenomenon in which the photochemical oxidation of aromatic VOCs is accelerated by intense solar radiation, which facilitates the chemical transformation of Sec-BrC and $O_3$. Consequently, more SW radiation was

absorbed by the Sec-BrC DRF, resulting in relatively high SW Sec-BrC DRFs during the summer months in most regions (Fig.
12e). In contrast to Pri-BrC and BC, the correlation between the atmospheric SW Sec-BrC DRF and $O_3$ concentration was
positive in all regions. Water vapor was positively correlated with the atmospheric LW Sec-BrC DRF in most areas, similar to
Pri-BrC and BC.

## 4 Conclusions

BC and BrC can contribute significantly to climate change through radiative forcing. Minimal research exists on the climate
change of Sec-BrC in China. We modified the GEOS-Chem coupled with RRTMG model for simulating the effects of Pri-
BrC and Sec-BrC light absorption based on local emission inventory, and report the direct radiative forcings contributed by
LACs and the interactions with influencing factors with implications for climate warming. The results show that the model
captured the seasonal variations of BC, Pri-BrC, and Sec-BrC with the most pronounced variations in winter at mean values
of $0.87 \pm 0.60$, $0.96 \pm 0.68$ and $0.05 \pm 0.02$ μg m$^{-3}$, respectively. The positive annual mean atmospheric net DRFs for BC, Pri-
BrC, and Sec-BrC in China under clear-sky conditions imply that they can warm the atmosphere, with values of $1.848 \pm 1.098$,
$0.146 \pm 0.079$, and $0.022 \pm 0.008$ W m$^{-2}$, respectively. The influences of LACs concentration, AOD, AAOD on corresponding
shortwave DRF within the atmosphere were positive in most regions. For SSA and $O_3$, their correlations with atmospheric
shortwave DRF were positive for Sec-BrC but negative for BC and Pri-BrC in most regions. For surface albedo, the correlation
with atmospheric shortwave DRF was not as explicit as other factors, which indicates that the interactions between LACs DRF
and surface albedo were more complicated. The surface longwave DRFs for the LACs showed negative correlations with water
vapor in most areas. The highest atmospheric warming effect of LACs was observed in Central China, followed by East China,
owing to the high LACs concentrations, AOD, and AAOD and low surface albedo and $O_3$ concentration.

    It should be noted that our study had some limitations and uncertainties. First, the sparse distribution of fewer monitoring
stations in the western region may limit the ability to assess the model performance in this region. Second, the ideal spherical
shape of LACs adopted in our model may lead to an overestimation of SSA and an underestimation of LACs radiative forcings.
In order to produce more accurate SSA and LACs DRFs, an improved parameterization of LACs shape is needed in the future.
Additionally, the LACs mixing state, aging rate, size distribution, and AAE values can affect the LACs concentrations and
optical properties. Furthermore, the mixing state, aging rate, and size distribution in different regions vary with time and the
AAE values mainly depend on the emission sources and burning conditions. Owing to the lack of refined local schemes for
these parameters in China, we used the external mixing state, fixed aging time, log-normal size distribution, and empirical
AAE values in our simulation, which may introduce uncertainties into the subsequent estimation of LACs DRFs. Although
the enhancement of BC absorption by the coating was set to 1.5 in this study, it should be optimized in the future according to
the aging process. The refined local emission inventories and more accurate physical and optical parameters of LACs are
required in the subsequent study so as to reduce the uncertainties of LACs DRFs. Finally, the exploration of influencing factors
associated with LACs DRFs may not be sufficiently comprehensive, as other variables (e.g., solar zenith angle and relative

humidity) can also affect LACs DRFs. Future studies should consider incorporating the effects of other variables on LACs DRFs and analyzing their interaction mechanisms with greater insight to provide a more realistic and comprehensive characterization of their relationships, thereby improving the modeling accuracy of LACs DRFs and deepening our understanding of LACs' climatic effects.

**Data availability**

The original code for GEOS-Chem model version 14.0.0 can be downloaded at https://github.com/geoschem/geos-chem. The AERONET Level 1.5 and 2.0 products can be obtained from https://aeronet.gsfc.nasa.gov/new_web/index.html. The reanalysis data of MERRA-2 and CAMS are available at https://disc.gsfc.nasa.gov/datasets/M2T1NXAER_5.12.4/summary, and https://ads.atmosphere.copernicus.eu/datasets/cams-global-reanalysis-eac4?tab=overview, respectively. The tropospheric column $O_3$ concentration can be accessed online at the following link of https://acd-ext.gsfc.nasa.gov/Data_services/cloud_slice/new_data.html. The surface albedo and water vapor content can be acquired from https://cds.climate.copernicus.eu/datasets/reanalysis-era5-land-monthly-means?tab=overview, and https://ladsweb.modaps.eosdis.nasa.gov/archive/allData/61/MOD08_M3, respectively.

**Author contributions**

LC and NC designed the project. NC and SY designed the model experiments. SY carried out the model experiments and prepared the paper. All the co-authors provided comments and reviewed the manuscript.

**Competing interests**

The contact author has declared that none of the authors has any competing interests.

**Disclaimer**

Publisher's note: Copernicus Publications remains neutral with regard to jurisdictional claims made in the text, published maps, institutional affiliations, or any other geographical representation in this paper. While Copernicus Publications makes every effort to include appropriate place names, the final responsibility lies with the authors.

**Acknowledgement**

This study was funded by the National Natural Science Foundation of China (grant no. 42477106) and the Natural Science Foundation of Tianjin (grant no. 22JCYBJC00150).

**Financial support**

This research has been supported by the National Natural Science Foundation of China (grant no. 42477106) and the Natural Science Foundation of Tianjin (grant no. 22JCYBJC00150).

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
