# Peer review of "Direct radiative forcing of light-absorbing carbonaceous aerosol and the influencing factors over China"

_EGUsphere, 2024_

## Author Response (AR3)

**Dear Editor:**

Thank you for your letter and for the comments concerning our manuscript entitled "Direct radiative forcing of light-absorbing carbonaceous aerosol and the influencing factors over China". Those comments are all valuable and very helpful for revising and improving our paper. We studied the comments carefully and made corrections which we hope meet with approval. Revised portions are marked in the revised manuscript with changes marked. The additions are shown in red, and the deletions are shown in grey and marked with strike-through. We also submit a revised manuscript with no changes marked. The responses to the editor's comments are as follows:

**Respond to Editor:**

Public justification (visible to the public if the article is accepted and published):
Dear authors,

Thanks for your going through the review process of your manuscript. Thanks to the reviewers for their valuable contributions.
As you have addressed all the points raised during the revision process, your manuscript is about being accepted. However, the abstract need some more editions to fit the guidelines of ACP manuscript. Please make sure that your abstract has at most 250 words in total. Please note that all acronyms use in the abstract must be properly defined at the first use, as the abstract must stand alone and completely comprehensible and understandable.
Once these two minor points will be handled in the sufficient way, we will be able to proceed further.

Regards

1. **Please make sure that your abstract has at most 250 words in total.**
   Response: Thanks for the careful checking. To ensure that the abstract section does not exceed 250 words, we revised this section as follows:

   Black carbon (BC) and brown carbon (BrC) are the dominant light-absorbing carbonaceous aerosols (LACs) that contribute significantly to climate change through absorbing and scattering radiation. We used GEOS-Chem integrated with Rapid Radiative Transfer Model for General Circulation Models to estimate LACs properties and direct radiative forcings (DRFs) in China. Primary BrC (Pri-BrC) and secondary BrC (Sec-BrC) were separated from organic carbon and modeled as independent tracers. LACs Chinese anthropogenic emissions and refractive

indexes were updated. Additionally, we investigated the impacts of LACs properties and atmospheric variables on LACs DRFs based on principal component analysis. It showed that BC exerts a warming effect at the top of the atmosphere, while Pri-BrC and Sec-BrC induce a cooling effect. At the surface, they collectively lead to surface cooling, whereas within the atmosphere, they all can contribute to atmospheric heating with $1.848 \pm 1.098$, $0.146 \pm 0.079$, and $0.022 \pm 0.008$ W m$^{-2}$, respectively. The atmospheric shortwave DRFs of BC and Pri-BrC were proportional to their corresponding concentrations, aerosol optical depth (AOD), and absorption aerosol optical depth (AAOD), and inversely proportional to single scattering albedo, surface albedo, and ozone concentration in most regions. The surface longwave DRFs for the LACs showed negative correlations with water vapor in most areas. The highest atmospheric warming effect of LACs was observed in Central China, followed by East China, owing to the high LACs concentrations, AOD, and AAOD and low surface albedo and ozone concentration. This study enhances our understanding of the climatic impacts of LACs.

2. **Please note that all acronyms use in the abstract must be properly defined at the first use, as the abstract must stand alone and completely comprehensible and understandable.**
   Response: Thank you for the kind reminder. We have ensured that all acronyms used in the abstract are properly defined at the first time they are used, please refer to the abstract section.

**Dear Editor and Referees:**

Thank you for your letter and for the thoughtful comments concerning our manuscript entitled "Direct radiative forcing of light-absorbing carbonaceous aerosol and the influencing factors over China". Those comments are all valuable and very helpful for revising and improving our paper. We studied the comments carefully and made corrections which we hope meet with approval. Revised portions are marked in the revised manuscript with changes marked. The additions are shown in red, and the deletions are shown in grey and marked with strike-through. We also submit a revised manuscript with no changes marked. The response to the editor and referee's comments are as follows:

**Respond to Referee #1:**

This is a quite complete paper about the radiative forcing due to carbonaceous aerosols in China. The selected period extended from October 2016 to December 2017. Surface data were obtained from 36 monitoring sites. Although gridded data were also used. The analysis considers a radiative model. Organic carbon is divided in primary and secondary brown carbon, the radiative forcing of both tracers is investigated together with the influence of different atmospheric variables. Since the analysis is quite detailed and focused, only some few minor changes are suggested.

1. **Most of the surface stations are placed at east of the country. The authors should comment the influence of such distribution on the results and if different results could be obtained with a homogeneous distribution of surface stations**
   Response: Thanks for the comments. In our study, the data collected at the monitoring stations were used to validate and evaluate the model performance in simulating the concentrations of light-absorbing carbonaceous aerosols. The 36 monitoring stations included in our manuscript cover most provinces across China. These sites are mainly distributed in central and eastern China, regions characterized by intense human activities, while they are relatively sparse in the less populated western part of China. The sparse distribution of fewer monitoring stations in the western region may limit us to assess the model performance in this region. We added this comment into the latter part of the conclusion section (Sect. 4) in our manuscript as follows:

   "It should be noted that our study had some limitations and uncertainties. First, the sparse distribution of fewer monitoring stations in western China may limit the ability to assess the model performance in this region. …"

2. **Simulation was considered from October 2016 to December 2017. The authors should introduce a comment about the result representativeness since meteorological conditions may be quite different in different years. Perhaps the model is not so sensitive against meteorological conditions, or the weight of weather variables is weak.**

Response: Thank you for the comments. We agree with your opinion that meteorological conditions may be different in different years, and the model can be affected by meteorological conditions or weather variables. The monitoring BC concentrations were collected from January to August 2017 in our study, and our simulation was conducted from October 2016 to December 2017 in order to compare with the monitoring data. The initial three months (October to December 2016) simulation was designated as the spin-up period to eliminate the influence of the initial conditions and stabilize the model.

3. **The uncertainty of radiative forcing is quite important. Perhaps, authors could explain the reason for such uncertainty and if they consider procedures to make it smaller or cases where this uncertainty is small.**
   Response: Thank you for the suggestions. The uncertainty of the radiative forcing is indeed important. The radiative forcings of light-absorbing carbonaceous aerosols can be affected by the emission inventory. To simulate the radiative forcing of light-absorbing carbonaceous aerosols, anthropogenic emissions over China were updated by the localized emission inventory INTAC with a resolution of $0.1° \times 0.1°$. However, emissions from biomass burning were derived from the global open biomass burning inventory GFED-4 with a resolution of $0.25° \times 0.25°$, which may introduce uncertainties in the radiative forcing of light-absorbing carbonaceous aerosols due to its relatively low resolution. Additionally, the physical and optical properties of light-absorbing carbonaceous aerosols vary dynamically in different regions, periods, and pollution conditions. However, the use of default values in the parameterization of these parameters may bring uncertainties into the radiative forcing estimation of light-absorbing carbonaceous aerosols. Future studies are expected to acquire refined local emission inventories and more accurate physical and optical parameters of light-absorbing carbonaceous aerosols, thereby reducing uncertainties in their radiative forcing. The uncertainties associated with the radiative forcing estimates of light-absorbing carbonaceous aerosols were mentioned in the latter part of the conclusion section (Sect. 4) in our manuscript. We modified this part after careful consideration. As follows:

"… Second, the LACs mixing state, aging rate, and AAE values can affect the LACs concentrations and optical properties. Furthermore, the mixing state and aging rate in different regions vary with time and the AAE values mainly depend on the emission sources and burning conditions. Owing to the lack of refined local schemes for these parameters in China, we used the external mixing state, fixed aging time, and empirical AAE values in our simulation, which may introduce uncertainties into the subsequent estimation of LACs DRFs. The refined local emission inventories and more accurate physical and optical parameters of LACs are acquired in the subsequent study to reduce the uncertainties of LACs DRFs. …"

**Respond to Referee #2:**

In this study, the authors investigate the radiative properties and direct radiative forcing of light absorbing carbonaceous aerosol over China using GEOS-Chem and the Rapid Radiative Transfer Model. A particular emphasis was placed on describing primary and secondary brown carbon; for this, the authors separated these species from organic carbon and modeled them as independent tracers. There are a lot of details involved in configuring the simulations. The choices that the authors made seem quite reasonable. The manuscript is well written, and the results are important. As such, in my opinion, this manuscript is appropriate for publication in Atmospheric Chemistry and Physics pending some clarifications and small corrections, as I list below.

1. **lines 44-45: If Xu et al. (2024) estimated the DRF of Sec-BrC in China, the authors should make it clearer what all the differences are between Xu et al.'s (2024) approach and the present approach. Is the only difference the global emission inventory?**
   Response: Thanks for the comments. We made careful comparisons with the methodology used in the study of Xu et al. 2024. First, the CAM5 model used by Xu et al. (2024) has a horizontal resolution of $0.9° \times 1.25°$ with 30 vertical layers, while the GEOS-Chem model we used has a horizontal resolution of $0.5° \times 0.625°$ with 47 vertical layers. Some studies showed that coarse-grid chemistry and transport models may not be able to accurately calculate aerosol concentrations because they are not able to reproduce the gradient of concentrations within a grid-box, especially when the mixing with clean air occurs (Rastigejev et al., 2010; Vignati et al., 2010). Xu et al. (2024) used a global emission inventory CEDS (Community Emission Data System) with a resolution of $0.5° \times 0.5°$ for anthropogenic emissions of BrC in China, whereas we updated this with a localized high-resolution inventory INTAC (High-Resolution Integrated Emission Inventory of Air Pollutants for China, $0.1° \times 0.1°$). They applied the same ratio of BrC emission for all fuel types, while in our study, we estimated sector-specific BrC emission ratios based on the relationship between modified combustion efficiency and aerosol light absorption. Second, Xu et al. (2024) only estimated the direct radiative forcing (DRF) of BrC at the top of the atmosphere. In order to better evaluate the climatic impacts of LACs, we simultaneously estimated DRFs of three components of LACs (BC, primary BrC, and secondary BrC) at the top of the atmosphere, within the atmosphere, and at the surface for LW, SW, and LW + SW (NET). Third, we further explored the effects of LACs properties and atmospheric variables on their radiative forcings. We improved the related descriptions as follows:

   "Although Xu et al. (2024) estimated the DRF of Sec-BrC in China, their simulation was based on the global emission inventory with a resolution of $0.5° \times 0.5°$, and a unified BrC emission ratio was applied to all fuel types. The local emission inventory with higher resolution and more accurate model parameters should be improved."

   Response: Thanks for the careful checking. We added the introductory words before Eqs. (5) and (6), and modified this paragraph as follows:

"By assuming the aerosol absorption is mainly contributed to BC and BrC, the separation of light absorption into BC and BrC follows the absorption Ångström exponent (AAE) segregation method, simply expressed by Eq. (5) and Eq. (6) as follows:

$$Abs_{BC}(\lambda) = Abs(880nm) \times \left(\frac{\lambda}{880}\right)^{-AAE_{BC}} \tag{5}$$

$$Abs_{BrC}(\lambda) = Abs(\lambda) - Abs_{BC}(\lambda) \tag{6}$$

where $Abs_{BC}(\lambda)$ and $Abs_{BrC}(\lambda)$ are the light absorption coefficients of BC and BrC at a given wavelength, respectively. AAE indicates the wavelength dependence of aerosol and can be obtained by the fit of multiwavelength absorption using the Power Law (Cao et al., 2024; Weingartner et al., 2003). The value of $AAE_{BC}$ is assumed to be 1.0, indicating "weak" spectral dependence of light absorption (Lack and Langridge, 2013; Cao et al., 2024). The light absorption coefficients for Pri-BrC and Sec-BrC are determined by the minimum R-squared approach, which is a BC-tracer method developed by Wang et al. (2019). More specific implementations of determinations for $Abs_{Pri-BrC}$ and $Abs_{Sec-BrC}$ can be found in Cao et al. (2024)."

3. **lines 108-109: "indicating BC is wavelength independent within 880 nm" – Is BC assumed to be wavelength independent over a range of wavelengths? If so, this wording is not clear.**
   Response: Thank you for the comments. In order to convey the meaning more clearly, we rephrased this sentence as follows:

"The value of $AAE_{BC}$ is assumed to be 1.0, indicating "weak" spectral dependence of light absorption (Lack and Langridge, 2013; Cao et al., 2024)."

4. **lines 141-142: "the entire China" – "the entirety of China"**
   Response: Thanks for the suggestions. We revised this sentence as follows:

   "Our simulation was performed at a 0.5° × 0.625° horizontal resolution in the nested domain of East Asia (60–150°E and 11°S–55°N), which covered the entirety of China; 47 hybrid sigma vertical layers stretching from the ground to the top of the modeled atmosphere (0.01 hPa, 80 km altitude), with the lowest level at about 60 m."

5. **lines 167-170: The authors should discuss the accuracy of using a standard Mie code for LAC, for which the particle shape is often not spherical. How do their single scattering results compare with results obtained in other studies accounting for the actual shape of such particles?**
   Response: Thank you for the comments. BC particles are aggregates of primary spheres (i.e., monomers) and typically have diameters of a few tens of nanometers (Tian et al., 2006; Adachi et al., 2010). Most climate and atmospheric chemistry transport models assume that carbonaceous aerosols are spheres, and their optical properties are calculated using Mie theory based on the spherical assumption, such as GEOS-Chem model. The ideal spherical shape of LACs adopted in our study may lead to an overestimation of SSA by approximately 19-25% (Adachi et al., 2011; China et al., 2015), thus resulting in an underestimation of LACs DRFs. In order to produce more accurate SSA and reduce uncertainties in radiative forcing estimates, an improved parameterization of LACs shape is needed. We added this comment to Section 3.2 (Climate change contribution by BC and interactions with influencing factors) and Section 4 (Conclusion) in our manuscript. As follows:

[revised manuscript text omitted]

Tian, K., Thomson, K. A., Liu, F., Snelling, D. R., Smallwood, G. J., and Wang, D.: Determination of the morphology of soot aggregates using the relative optical density method for the analysis of TEM images, Combust. Flame, 144, 782–791, https://doi.org/10.1016/j.combustflame.2005.06.017, 2006.

6.  **line 175: Are all of the values of DRF from every 3-hour time step used to calculate the annual mean DRF? If so, is the average a simple average or a weighted average?**
    Response: Thank you for the comments. RRTMG calculates LACs DRFs at a 3-hour time step and the results were converted to a daily average within the model. The annual average DRF is calculated by the daily data, not a weighted average. We added this information to our manuscript as follows:

    "RRTMG calculates atmospheric radiation in the wavelengths between 0.23 and 56 μm, which can be further divided into 16 longwave (LW) bands and 14 shortwave (SW) bands (Iacono et al., 2008; Heald et al., 2014; Methymaki et al., 2023). It was called every 3 h to compute the LACs DRFs at the top of the atmosphere (TOA), in the atmosphere (ATM), and at the surface (SUR) for LW, SW, and LW + SW (NET). The DRFs we output were at a daily time step, which were further averaged to obtain the annual mean values. The total flux and flux without a particular species can be calculated separately and the difference between the two fluxes is considered the DRF of interested species (Heald et al., 2014; Wang et al., 2014). Hence, the definition of aerosol DRF is as follows:"

7.  **line 188: "considering that the fraction" – "considering that a fraction"**
    Response: Thanks for the careful checking. We revised the sentence as follows:

    "Considering that a fraction of primary OC is BrC, we derived the Pri-BrC emissions following the methodology presented by Jo et al. (2016), which is summarized as follows:"

8.  **line 207: "others k values" – "other k values"**
    Response: Thanks for the careful checking. We corrected the spelling as follows:

    "We calculated the primary $k_{BrC,550}$ using Eq. (4) derived from Sun et al. (2007), based on $MAE_{BrC,550} = 0.886\ m^2\ g^{-1}$ used in BrC emission calculation; other k values were derived from Eq. (5) obtained from Saleh et al. (2014)."

9.  **line 208: A period is missing after "Saleh et al. (2014)".**
    Response: Thanks for the careful checking. We revised the sentence as follows:

    "We calculated the primary $k_{BrC,550}$ using Eq. (4) derived from Sun et al. (2007), based on $MAE_{BrC,550} = 0.886\ m^2\ g^{-1}$ used in BrC emission calculation; other k values were derived from Eq. (5) obtained from Saleh et al. (2014)."

10. **lines 488-197: The authors discuss some limitations of the study, such as their choice of external mixing state,**

**fixed aging time, and empirical AAE. The authors should discuss whether changing any of the other details from section 2.3 regarding the configuration of the simulations to other reasonable options could change their results in any significant way.**

Response: Thanks for the comments. Apart from the external mixing state, fixed aging time, and empirical AAE, we discussed the effects of other parameters. The freshly emitted BC is externally mixed and consists of aggregated primary spherules (i.e., monomers), usually having a diameter of tens of nanometers (Adachi et al., 2010; Tian et al., 2006; Tuccella et al., 2020). During atmospheric transport, BC undergoes various aging processes such as coagulation, condensation, and heterogeneous reactions, resulting in changes in its morphology (size, shape, and internal structure) and mixing state (e.g., soot coating) (China et al., 2015; Scarnato et al., 2013). When BC is coated with non-absorbing materials, its absorption ability can be amplified and this increase can affect the total climate forcing (Bond et al., 2006). Bond et al. (2006) recommended that absorption by aged BC is about 1.5 times greater than that of fresh BC. Thus, the BC absorption enhancement factor of 1.5 was used in our study and it should be optimized according to the aging processes in the future to obtain more accurate radiative forcings of LACs. Curci et al. (2019) found a slight increase in SSA in the externally mixed case by increasing BC radius to account for its aging processes, and BC particle size would be unlikely to significantly affect its radiative forcing. Previous studies have shown that the BC spherical shape used in Mie theory may overestimate light scattering by 19-25% (China et al., 2015; Adachi et al., 2011). Therefore, the assumption of spherical shape for LACs in the standard GEOS-Chem model may lead to an overestimation of SSA and an underestimation of their radiative forcings. In order to produce more accurate SSA and reduce uncertainties in radiative forcing estimates, an improved parameterization of LACs shape is needed. We added this comment to Section Section 3.2 (Climate change contribution by BC and interactions with influencing factors) and Section 4 (Conclusion) in our manuscript. As follows:

"Figure 6 shows the spatial distribution of the simulated annual mean clear-sky BC DRF in China. The values of the BC DRFs for all-sky conditions (Fig. S4) were generally smaller than those for clear-sky conditions, which agrees with previous findings (Feng et al., 2013; Heald et al., 2014; Lin et al., 2014). This can be primarily due to the scattering by clouds masking the scattering by aerosols. The mean clear-sky BC DRFs across China for LW, SW, and NET at the TOA were $0.004 \pm 0.002$, $0.381 \pm 0.238$, and $0.385 \pm 0.240$ W m$^{-2}$, respectively, whereas the surface mean values were $0.025 \pm 0.017$, $-1.487 \pm 0.883$, and $-1.462 \pm 0.867$ W m$^{-2}$, respectively. A positive NET BC DRF at the TOA indicates a net heating effect on the top of the atmosphere, whereas a negative value at the surface represents a net cooling effect on the ground. The positive NET BC DRF within the atmosphere with a mean value of $1.848 \pm 1.098$ W m$^{-2}$ indicates that BC can strongly absorb solar radiation, leading to atmospheric warming. BC warming effect is likely to be underestimated due to the ideal spherical shape of LACs adopted in our model, which may lead to an overestimation of SSA by approximately 19-25% (China et al., 2015; Adachi et al., 2011)."

"It should be noted that our study had some limitations and uncertainties. First, the sparse distribution of fewer monitoring stations in the western region may limit the ability to assess the model performance in this region. Second, the ideal spherical shape of LACs adopted in our model may lead to an overestimation of SSA and an underestimation of LACs radiative forcings. In order to produce more accurate SSA and LACs DRFs, an improved parameterization of LACs shape is needed in the future. Additionally, the LACs mixing state, aging rate, size distribution, and AAE values can affect the LACs concentrations and optical properties. Furthermore, the mixing state, aging rate, and size distribution in different regions vary with time and the AAE values mainly depend on the emission sources and burning conditions. Owing to the lack of refined local schemes for these parameters in China, we used the external mixing state, fixed aging time, log-normal size distribution, and empirical AAE values in our simulation, which may introduce uncertainties into the subsequent estimation of LACs DRFs. Although the enhancement of BC absorption by the coating was set to 1.5 in this study, it should be optimized in the future according to the aging process. The refined local emission inventories and more accurate physical and optical parameters of LACs are required in the subsequent study so as to reduce the uncertainties of LACs DRFs. Finally, the exploration of influencing factors associated with LACs DRFs may not be sufficiently comprehensive, as other variables (e.g., solar zenith angle and relative humidity) can also affect LACs DRFs. Future studies should consider incorporating the effects of other variables on LACs DRFs and analyzing their interaction mechanisms with greater insight to provide a more realistic and comprehensive characterization of their relationships, thereby improving the modeling accuracy of LACs DRFs and deepening our understanding of LACs' climatic effects."